# Eicosapentaenoic acid induces macrophage Mox polarization to prevent diabetic cardiomyopathy

Jie Li[1,2,8], Wenshan Nan [1,3,8], Xiaoli Huang[4], Huali Meng[1,2], Shue Wang[5], Yan Zheng[1], Ying Li[1], Hui Li[6], Zhiyue Zhang[6], Lei Du [1,2], Xiao Yin [3✉] & Hao Wu [1,2,7✉]

## Abstract

**Diabetic cardiomyopathy (DC) leads to heart failure, with few effective approaches for its intervention. Eicosapentaenoic acid (EPA) is an essential nutrient that benefits the cardiovascular system, but its effect on DC remains unknown. Here, we report that EPA protects against DC in streptozotocin and high-fat diet-induced diabetic mice, with an emphasis on the reduction of cardiac M1-polarized macrophages. In vitro, EPA abrogates cardiomyocyte injury induced by M1-polarized macrophages, switching macrophage phenotype from M1 to Mox, but not M2, polarization. Moreover, macrophage Mox polarization combats M1-polarized macrophage-induced cardiomyocyte injury. Further, heme oxygenase 1 (HO-1) was identified to maintain the Mox phenotype, mediating EPA suppression of macrophage M1 polarization and the consequential cardiomyocyte injury. Mechanistic studies reveal that G-protein-coupled receptor 120 mediates the upregulation of HO-1 by EPA. Notably, EPA promotes Mox polarization in monocyte-derived macrophages from diabetic patients. The current study provides EPA and macrophage Mox polarization as novel strategies for DC intervention.**

**Keywords** Diabetic Cardiomyopathy; Eicosapentaenoic Acid; Heme Oxygenase 1; Macrophage; Mox Polarization
**Subject Categories** Immunology; Metabolism; Molecular Biology of Disease

## Introduction

Diabetes mellitus (DM) poses a severe threat to public health and social economy worldwide. In 2021, International Diabetes Federation reported that 536.6 million adults were affected by DM globally (Yang and Yang, 2022), and this number was predicted to rise up to 783.2 million by 2045 (Sun et al, 2022a). According to the World Health Organization, DM was the ninth cause of global death in 2019, with cardiovascular complications being the major cause of death (da Silva et al, 2022).

Diabetic cardiomyopathy (DC), a common complication of DM, is defined as myocardial dysfunction that develops from complex pathophysiological mechanisms of DM in the absence of coronary artery disease or hypertension (Guo et al, 2022). Epidemiological studies have identified a high incidence (19–26%) of heart failure in patients with DM (Jia et al, 2018a). Despite the rapid increased number of preclinical and clinical studies on DC in the past decades, the pathogenesis of DC is still not well-known. As a result, there still lacks effective preventive or therapeutic approaches for DC intervention (Zhao et al, 2022).

Macrophages play important roles in the pathogenesis and progression of DC (Bajpai and Tilley, 2018). M1 polarization of macrophages produces pro-inflammatory cytokines that are involved in diabetic heart injury (Bajpai and Tilley, 2018; Elmadbouh and Singla, 2021). Conversely, M2 polarization of macrophages reduces cardiac inflammation under experimental conditions of DM (Jadhav et al, 2013). In addition to M1 and M2, a spectrum of macrophage phenotypes has been identified, playing key roles in various physiological and pathophysiological processes (Xue et al, 2014). Among these phenotypes, the oxidized phospholipids-induced macrophages (Mox), found in 2010 by Kadl and colleagues, has drawn attention in recent years (Kadl et al, 2010). This phenotype possesses anti-inflammatory and antioxidant effects that prevent disease progression (Bonetti et al, 2021; Eren et al, 2018; Li et al, 2022). However, the role of Mox in DC was not previously known.

Omega-3 polyunsaturated fatty acids (n-3 PUFAs), especially eicosapentaenoic acid (EPA) and docosahexaenoic acid (DHA), are linked to or even directly confer significant health advantages to patients with type 2 DM (T2DM) (Kumar et al, 2022). Supplementation of DHA and EPA attenuated hyperglycemia and insulin resistance without changing body weight in db/db mice, with EPA showing a more profound effect (Zhuang et al, 2021).

[1]Research Center of Translational Medicine, Jinan Central Hospital, Shandong University, 105 Jiefang Rd., Jinan, Shandong 250013, China. [2]Department of Nutrition and Food Hygiene, School of Public Health, Cheeloo College of Medicine, Shandong University, 44 Wenhua Xi Rd., Jinan, Shandong 250012, China. [3]Department of Endocrinology and Metabolism, Central Hospital Affiliated to Shandong First Medical University, Shandong First Medical University, 105 Jiefang Rd., Jinan, Shandong 250013, China. [4]Department of Nutrition, Qilu Hospital, Cheeloo College of Medicine, Shandong University, 107 Wenhua Xi Rd., Jinan, Shandong 250012, China. [5]Experimental Center of Public Health and Preventive Medicine, School of Public Health, Cheeloo College of Medicine, Shandong University, 44 Wenhua Xi Rd., Jinan, Shandong 250012, China. [6]NMPA Key Laboratory for Technology Research and Evaluation of Drug Products, Key Laboratory of Chemical Biology (Ministry of Education), Department of Pharmaceutics, School of Pharmaceutical Sciences, Cheeloo College of Medicine, Shandong University, 44 Wenhua Xi Rd., Jinan, Shandong 250012, China. [7]Shandong Provincial Engineering and Technology Research Center for Food Safety Monitoring and Evaluation, 44 Wenhua Xi Rd., Jinan, Shandong 250012, China. [8]These authors contributed equally: Jie Li, Wenshan Nan. ✉E-mail: yinxiao@sdu.edu.cn; hwu@sdu.edu.cn

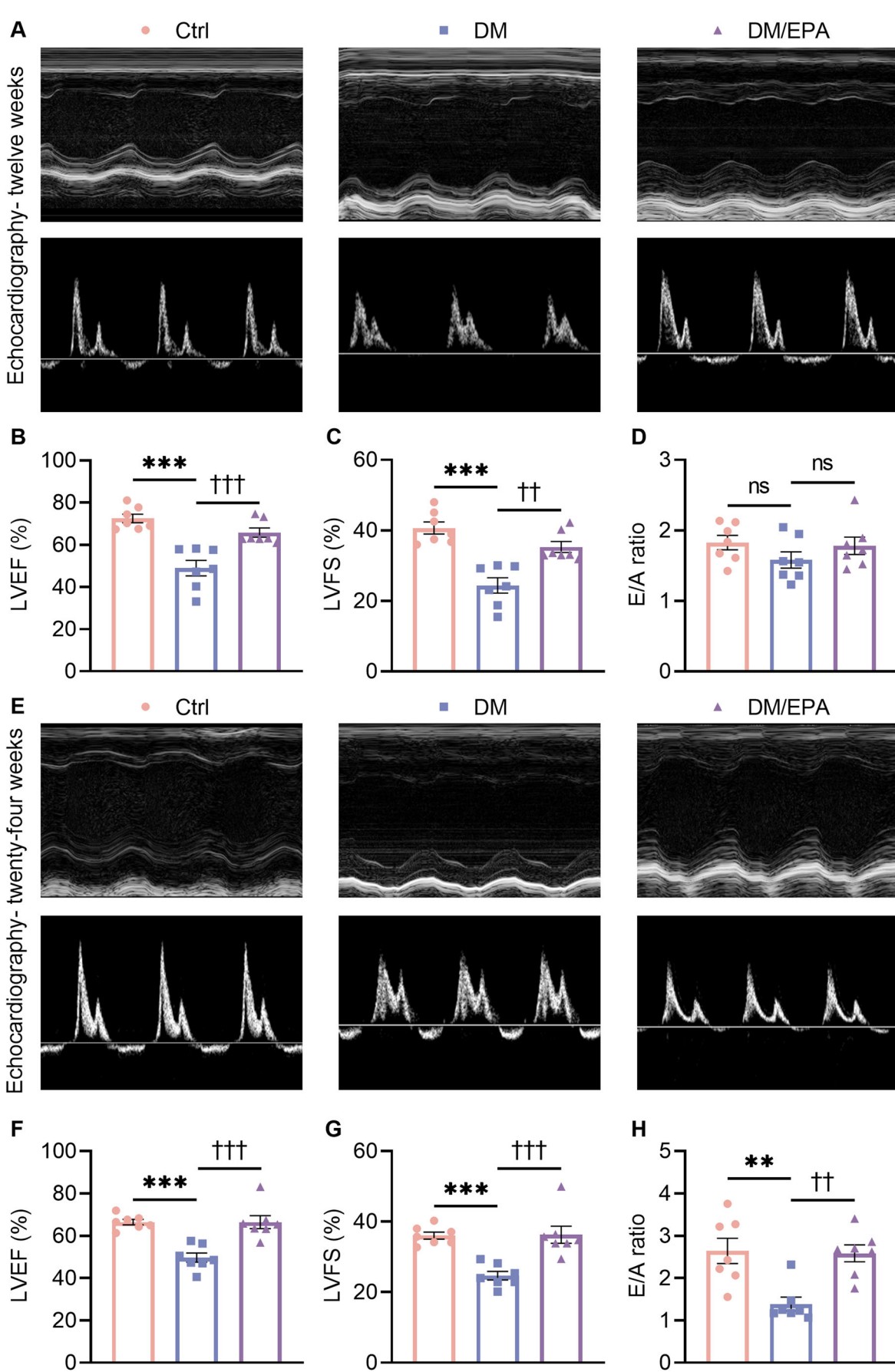

**Figure 1.  EPA improved cardiac dysfunction in the diabetic mice.**

(A) Representative transthoracic echocardiography M-mode and Doppler images and (B–D) LVEF, LVFS, and E/A ratio by echocardiographic detection twelve weeks post DM onset. For (B), ***$P < 0.001$, DM vs. Ctrl; †††$P < 0.001$, DM/EPA vs. DM. For (C), ***$P < 0.001$, DM vs. Ctrl; ††$P = 0.001$, DM/EPA vs. DM. (E) Representative transthoracic echocardiography M-mode and Doppler images and (F–H) LVEF, LVFS and E/A ratio by echocardiographic detection twenty-four weeks post DM onset. For (F), ***$P < 0.001$, DM vs. Ctrl; †††$P < 0.001$, DM/EPA vs. DM. For (G), ***$P < 0.001$, DM vs. Ctrl; †††$P < 0.001$, DM/EPA vs. DM. For (H), **$P = 0.001$, DM vs. Ctrl; ††$P = 0.002$, DM/EPA vs. DM. Data information: Data are represented as individual data points of $n = 7$ (B–D, F–H) biological replicates and means ± SEM. ns, not significant. Analysis by one-way ANOVA. EPA eicosapentaenoic acid, E/A mitral peak velocity ratio of early-diastolic filling to atrial contraction, LVEF left ventricular ejection fraction, LVFS left ventricular fraction shortening. Groups: Ctrl control, DM diabetes mellitus, DM/EPA diabetic mice supplemented with EPA. Source data are available online for this figure.

Moreover, large-scale clinical trials of n-3 PUFAs found reduction of cardiovascular events and mortality, as well as improved left ventricular structure and function in patients with heart failure (Siscovick et al, 2017; Yokoyama et al, 2007). However, the effect of EPA on DC and the underlying mechanism remain unclear.

EPA exerts antioxidant and anti-inflammatory activities (Djuricic and Calder, 2021). Notably, EPA has been shown to alter macrophage phenotypes. In a rat model of spontaneous hypertension, EPA was found to enhance M2 polarization in the heart, reducing cardiac inflammation and fibrosis (Gharraee et al, 2022). In addition, EPA inhibited M1 polarization, protecting against myocardial infarction induced by left coronary artery ligation in mice (Takamura et al, 2017). These studies indicated that macrophage phenotype alteration might be an important mechanism through which EPA protects from cardiomyocyte injuries.

G protein-coupled receptors (GPRs) are important signaling molecules for many cellular functions. When bound by ligands, GPRs activate a variety of cellular responses via several second messenger pathways; e.g., modulation of cAMP production, the phospholipase C pathway, ion channels, and mitogen-activated protein kinases (Gether, 2000; Ulloa-Aguirre et al, 1999). It was reported that five orphan receptors, GPR40, GPR41, GPR43, GPR84, and GPR120, could be activated by free fatty acids. Among these GPRs, GPR40 and GPR120 could interact with long-chain fatty acids such as EPA and DHA (Hirasawa et al, 2005; Itoh et al, 2003). GPR40 and GPR120 were both required for EPA's inhibitory effect on acute cerebral infarction-induced inflammation in mice (Mo et al, 2020). Previous research reported that GPR120 was highly expressed in M1 polarized macrophages (Oh et al, 2010), and GPR120-mediated signaling pathways could be activated by EPA in macrophages (Han et al, 2017). Thus, we hypothesized that EPA might function through activating macrophage GPR40 and/or GPR120.

In summary, the present study aimed to investigate the effect of EPA on DC in a mouse model of T2DM. The molecular action of EPA was studied using M1-polarized macrophages, as well as co-cultured M1-polarized macrophages and cardiomyocytes. The effect and molecular action of EPA were further verified in macrophages from patients with T2DM.

# Results

## EPA lowered fasting blood glucose level, and improved glucose tolerance and insulin sensitivity in the diabetic mice

EPA was studied for its effect in streptozotocin (STZ) and high-fat diet (HFD)-induced mouse model of T2DM (Fig. EV1A). EPA decreased fasting blood glucose (FBG) levels starting from the 20th week post DM onset (Fig. EV1B). EPA led to a mild increase in body weight of the diabetic mice starting from the 18th week post DM onset (Fig. EV1C). Glucose tolerance test (GTT) and insulin tolerance test (ITT) were also improved by EPA, as evidenced by the EPA-decreased glucose levels at the relevant time points and areas under the blood glucose curves (Fig. EV1D–G).

## EPA improved cardiac function in the diabetic mice

EPA was tested for its effect on DM-induced cardiac dysfunction at the ends of 12 and 24 weeks of EPA supplementation. After a 12-week intervention, EPA reversed the DM-induced decline of left ventricular ejection fraction (LVEF) and left ventricular fractional shortening (LVFS), but had no impact on mitral peak velocity ratio of early-diastolic filling to atrial contraction (E/A ratio) (Fig. 1A–D). After a 24-week EPA supplementation, LVEF, LVFS, and E/A ratio were all significantly improved in the diabetic mice (Fig. 1E–H).

## EPA prevented the diabetes-induced cardiac pathological injuries

To investigate the effect of EPA on DM-induced cardiac pathological injuries, the hearts underwent histopathological assessments. EPA prevented the DM-induced cardiac enlargement as revealed by the gross images and H&E staining (Fig. EV2A). These effects were confirmed by the decreased heart-weight-to-tibia length ratio in the EPA-treated diabetic mice (Fig. EV2B). The EPA-induced heart weight loss might be owing to the attenuation of cardiac hypertrophy as shown by wheat germ agglutinin (WGA) staining (Fig. EV2C,D). Additionally, EPA remarkably reduced the infiltrated inflammatory cells into the paravascular spaces as shown by H&E staining (Fig. EV2E,F). The immunohistochemical staining of F4/80 showed increased macrophage infiltration in diabetic mice, which was reduced by EPA (Fig. EV2G,H). Masson's trichrome staining revealed that EPA ameliorated cardiac fibrosis in the diabetic mice (Fig. EV2I,J). Further, EPA reduced Terminal Deoxynucleotidyl Transferase (TdT)-Mediated dUTP Nick-End Labeling (TUNEL) positive cells, suggesting that EPA prevented the DM-induced cardiac cell death (Fig. EV2K,L).

## EPA alleviated the DM-induced cardiac inflammation and oxidative stress

Persistent chronic inflammation and oxidative stress contribute to the pathogenesis of DC (Jia et al, 2018b; Liu et al, 2021; Sun et al, 2022b).

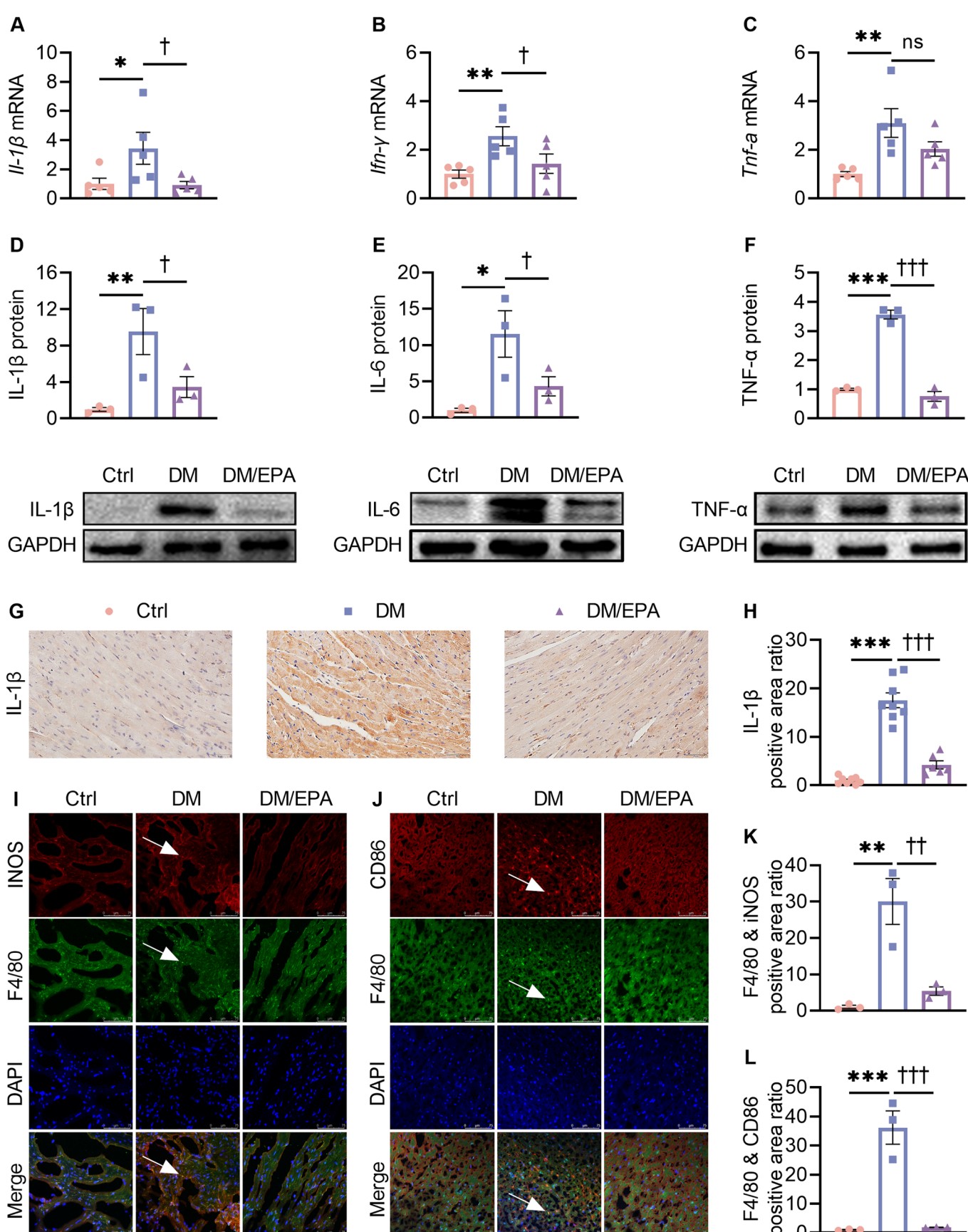

**Figure 2.  EPA alleviated DM-induced cardiac inflammation.**

(A–C) Cardiac mRNA expression of *Il-1β*, *Ifn-γ*, and *Tnf-α* were determined by qRT-PCR. For (A), *$P$ = 0.027, DM vs. Ctrl; †$P$ = 0.023, DM/EPA vs. DM. For (B), **$P$ = 0.007, DM vs. Ctrl; †$P$ = 0.036, DM/EPA vs. DM. For (C), **$P$ = 0.002, DM vs. Ctrl. (D–F) Cardiac protein expression of IL-1β, IL-6, and TNF-α were determined by Western blot. For (D), **$P$ = 0.009, DM vs. Ctrl; †$P$ = 0.036, DM/EPA vs. DM. For (E), *$P$ = 0.01, DM vs. Ctrl; †$P$ = 0.044, DM/EPA vs. DM. For (F), ***$P$ < 0.001, DM vs. Ctrl; †††$P$ < 0.001, DM/EPA vs. DM. (G, H) Immunohistochemical staining of IL-1β and quantification of positive area (bar = 20 μm). For (H), ***$P$ < 0.001, DM vs. Ctrl; †††$P$ < 0.001, DM/EPA vs. DM. (I) Immunofluorescence of iNOS (red) and F4/80 (green) (bar = 75 μm). The white arrows point to regions with iNOS and F4/80 fluorescence. (J) Immunofluorescence of CD86 (red) and F4/80 (green) (bar = 75 μm). The white arrows point to regions with CD86 and F4/80 fluorescence. (K, L) Quantification of merged immunofluorescence of F4/80 with either iNOS or CD86. For (K), **$P$ = 0.001, DM vs. Ctrl; ††$P$ = 0.003, DM/EPA vs. DM. For (L), ***$P$ < 0.001 DM vs. Ctrl; †††$P$ < 0.001, DM/EPA vs. DM. Data information: For (A–F, H, K, L), the data are normalized to Ctrl. Data are represented as individual data points of $n$ = 5, 5, 5 (A–C), $n$ = 3, 3 (D–F), $n$ = 8, 8, 6 (H), $n$ = 3, 3, 3 (K, L) biological replicates and means ± SEM. Analysis by one-way ANOVA. DAPI 4′,6-diamidino-2-phenylindole, *Ifn-γ* interferon gamma, *Il-1β*/IL-1β interleukin-1 beta, IL-6 interleukin-6, iNOS inducible nitric oxide synthase, *Tnf-α*/TNF-α tumor necrosis factor-α. Other abbreviations are the same as in Fig. 1. Groups: Ctrl control, DM diabetes mellitus, DM/EPA diabetic mice supplemented with EPA. Source data are available online for this figure.

Increased mRNA levels of interleukin-1 beta (*Il-1β*), interferon-γ (*Ifn-γ*), and tumor necrosis factor-α (*Tnf-α*), as well as protein levels of IL-1β, interleukin-6 (IL-6) and TNF-α were found in the hearts of the diabetic mice (Fig. 2A–H). EPA abolished these DM-induced effects (Fig. 2A,B,D–H), except for *Tnf-α* mRNA level (Fig. 2C, $P$ = 0.074). Notably, M1 polarized macrophages were accumulated in the diabetic hearts as evidenced by the enhancement of inducible nitric oxide synthase (iNOS) and F4/80 co-localization, as well as CD86 and F4/80 co-localization (Fig. 2I–L). This effect was not evident in EPA-treated diabetic hearts (Fig. 2I–L).

The following study assessed the effect of EPA on cardiac oxidative stress in the diabetic mice. EPA reduced the mRNA levels of NADPH oxidase 4 (*Nox4*) and nitric oxide synthase 2 (*Nos2*), and lowered the protein levels of NOX4 and cyclooxygenase-2 (COX2) (Fig. EV3A–D). Moreover, EPA reduced the positive stain of 8-hydroxy-2'-deoxyguanosine (8-OHdG) and reactive oxygen species (ROS) that were enhanced in the diabetic hearts (Fig. EV3E–H).

## EPA abrogated cardiomyocyte injury induced by M1 polarized macrophages

Since macrophage infiltration and M1 polarization contribute to cardiomyocyte injury under diabetic condition (Liu et al, 2021), and EPA was found to reduce M1 polarized macrophage infiltration into the diabetic hearts (Fig. 2I–L), we hypothesized that EPA might protect against DM-induced cardiac injury at least in part through inhibition of macrophage M1 polarization. Therefore, EPA was studied for its effect on M1 polarized macrophage-induced cardiomyocyte injury, using a co-culture model of HL-1 cells and lipopolysaccharides (LPS)-stimulated RAW264.7 cells (Fig. 3A).

RNA-sequencing (RNA-seq) showed a remarkable EPA-induced alteration of global RNA expression in M1 polarized RAW264.7 cells-co-cultured HL-1 cells (Appendix Fig. S1), addressing immune or inflammation-related pathways (Appendix Fig. S2). Quantitative real-time PCR (qRT-PCR) and Western blot further confirmed the inhibitory effect of EPA on M1 polarized RAW264.7 cells-provoked mRNA expression of *Nos2*, *Il-6*, and C-C motif chemokine ligand 2 (*Ccl2*) (Fig. 3B–D), protein expression of IL-6, TNF-α, COX2, NOX4, and NADPH oxidase 2 (NOX2) (Fig. 3E–I), as well as ROS level (Fig. 3J,K). These results identified a crucial role of M1 polarized macrophages in cardiomyocyte inflammation and oxidative stress, and suggested that EPA might protect cardiomyocytes by modulating macrophage M1 polarization.

## EPA modulated M1 and Mox, but not M2, polarization of macrophages

To investigate the effect of EPA on M1 polarized macrophages, EPA- or DMSO-treated M1 polarized RAW264.7 cells underwent RNA-seq (Fig. EV4A) which showed a dramatic different global RNA expression pattern (Fig. EV4B). LPS changed the RNA expression of over 2000 genes compared with Ctrl (Fig. EV4C). EPA altered 392 genes compared with DMSO, whereas DMSO only lead to an alteration of 29 genes (Fig. EV4C). Gene Ontology biological process analysis revealed signaling pathways significantly modulated by EPA, most of which were inflammatory pathways (Fig. EV4D).

In order to investigate the effect of EPA on macrophage polarization, indicators for M1 and M2 polarizations were determined (Fig. 4A). EPA had a dramatic inhibitory effect on M1 polarization, as evidenced by its significant inhibitory effect on mRNA levels of *Il-6*, *Il-1β*, and *Nos2*, as well as protein levels of IL-6, IL-1β, and TNF-α (Fig. 4B–G). However, EPA did not produce a convincing effect on the activation of M2 indicators of M1 polarized macrophages, as shown by the mRNA levels of arginase 1 (*Arg1*), transforming growth factor-β1 (*Tgf-β1*) and interleukin-10 (*Il-10*) (Appendix Fig. S3A–C), and the protein levels of ARG1 and TGF-β1 (Appendix Fig. S3D,E). Moreover, EPA was found to have no significant effect on the M2 polarized macrophages in the diabetic hearts (Appendix Fig. S3F,G). Further research validated the effect of EPA on M2 polarization of macrophages in vitro (Appendix Fig. S4A). It was found that EPA significantly upregulated the levels of ARG1, TGF-β1, and IL-10 proteins (Appendix Fig. S4B–D). These results showed that although EPA promoted M2 polarization in vitro, it did not affect M2 polarization in the hearts of diabetic mice, nor did it induce the M1 to M2 transition.

Given that the effect of EPA on M2 polarization could not be an explanation for its inhibition of M1 polarization, the following studies further explored the action of EPA on macrophages. By employing volcano plot, heme oxygenase 1 (*Hmox1*) mRNA level was identified to be dramatically increased by EPA (Fig. 5A), as a sign of Mox polarization (Kadl et al, 2010). Further, qRT-PCR and Western blot analyses confirmed the positive effect of EPA on Mox polarization, as evidenced by the increased mRNA levels of *Hmox1*, thioredoxin reductase 1 (*TrxR1*) and sulfiredoxin-1 (*Srxn1*), and protein levels of Hmox1 (HO-1) and TrxR1 (Fig. 5B–F).

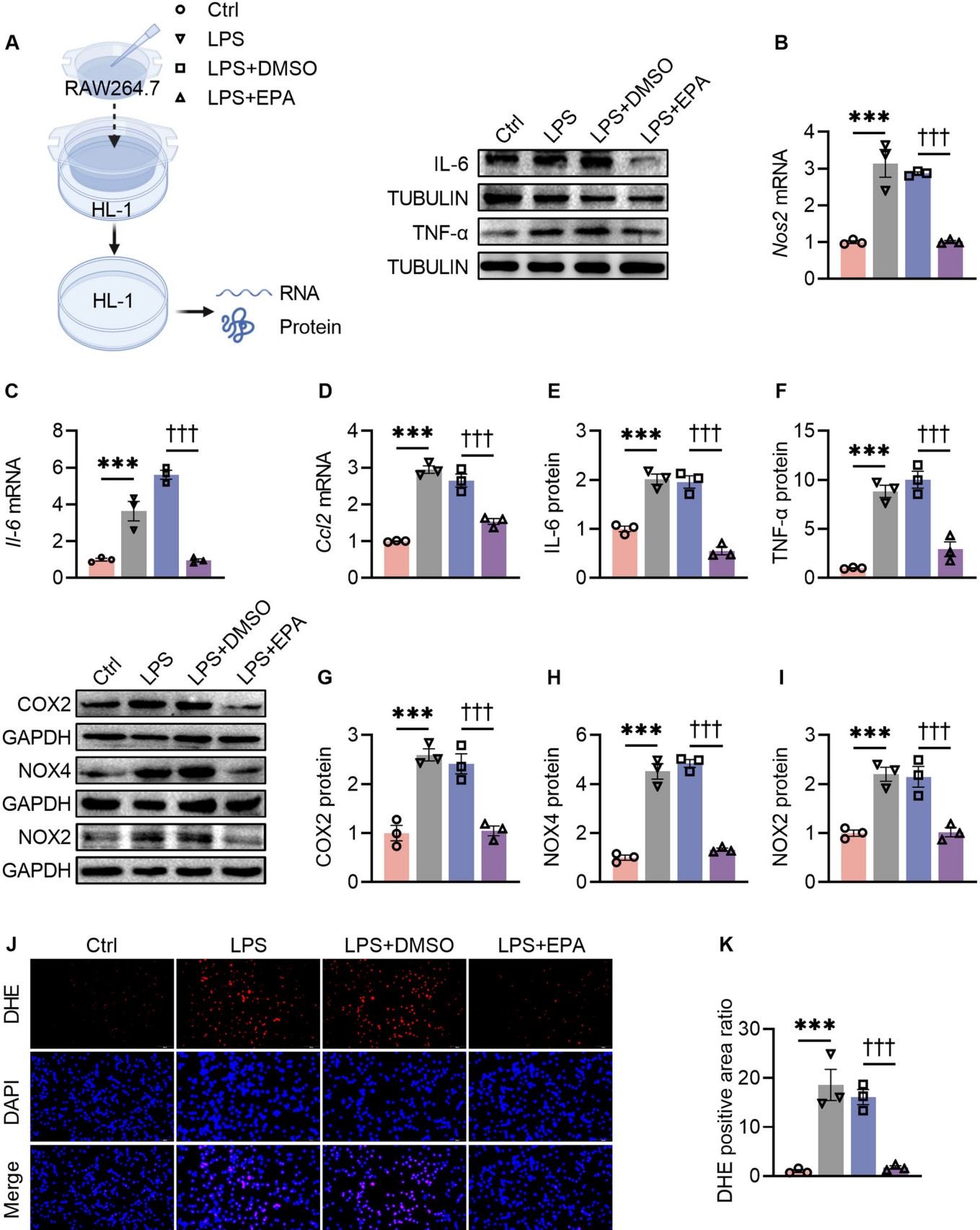

**Figure 3.   EPA abrogated cardiomyocyte injury induced by M1-polarized macrophages.**

(A) Schematic representation of the experimental protocol. (B–D) mRNA expression of *Nos2*, *Il-6*, and *Ccl2* were detected by qRT-PCR in HL-1 cells. ***$P < 0.001$, LPS vs. Ctrl; †††$P < 0.001$, LPS + EPA vs. LPS + DMSO. (E–I) IL-6, TNF-α, COX2, NOX4, and NOX2 protein levels in HL-1 cells determined by Western blot. ***$P < 0.001$, LPS vs. Ctrl; †††$P < 0.001$, LPS + EPA vs. LPS + DMSO (J, K) DHE assay (bar = 100 μm) by double staining with DAPI (blue) and DHE (red), and ratio of DHE positive cells. For (K), ***$P < 0.001$, LPS vs. Ctrl; †††$P < 0.001$, LPS + EPA vs. LPS + DMSO. Data information: For (B–I, K), the data are normalized to Ctrl. Data are represented as individual data points of $n = 3$ (B–I, K) biological replicates and means ± SEM. Analysis by one-way ANOVA. *Ccl2* C-C motif chemokine ligand 2, COX2 cyclooxygenase-2, DHE dihydroethidium, LPS lipopolysaccharides, *Nos2* nitric oxide synthase 2, NOX2/4 NADPH oxidase 2/4. Other abbreviations are the same as in Figs. 1 and 2. Groups: Ctrl control, LPS HL-1 cells co-cultured with LPS-stimulated RAW264.7 cells, LPS + DMSO HL-1 cells co-cultured with LPS-stimulated RAW264.7 cells and treated with DMSO, LPS + EPA HL-1 cells co-cultured with LPS-stimulated RAW264.7 cells and treated with EPA. Source data are available online for this figure.

## Macrophage Mox polarization combated M1 polarized macrophage-induced cardiomyocyte injury

Despite the EPA-induced switching of macrophage from M1 towards Mox polarization, it was unknown whether Mox polarization protected against M1 polarization-induced cardiomyocyte injury. Therefore, Mox polarization was established in RAW264.7 cells using oxidized 1-palmitoyl-2-arachidonoyl-sn-glycero-3-phosphocholine (ox-PAPC) (Appendix Fig. S5), followed by a 24-h incubation with HL-1 cells preliminarily co-cultured 24 h with M1 polarized RAW264.7 cells (Fig. 5G). Mox polarized RAW264.7 cells significantly decreased the protein levels of IL-6, TNF-α, COX2, and NOX4 in HL-1 cells, all of which were elevated by M1 polarized RAW264.7 cells (Fig. 5H–K).

## HO-1 maintained the Mox phenotype of macrophages, and played a critical role in EPA suppression of macrophage M1 polarization

As RNA-seq identified *Hmox1* mRNA was drastically elevated by EPA, and its protein product HO-1 is a hallmark of the macrophage Mox polarization, the following study researched whether HO-1 was required for the maintenance of Mox phenotype of macrophages using siRNA-induced gene silencing strategy (Appendix Fig. S6A,B). This resulted in decreased protein level of the Mox polarization hallmarks HO-1 and TrxR1 (Appendix Fig. S6C,D), indicating a crucial role of HO-1 in the maintenance of macrophage Mox phenotype.

The following study further researched whether HO-1 mediated EPA's inhibitory effect on macrophage M1 polarization by silencing *Hmox1* gene in LPS-stimulated RAW264.7 cells in the presence of EPA (Fig. 6A). The EPA-induced suppression of Il-6, Il-1β, and TNF-α expression were completely abrogated by *Hmox1* gene silencing (Fig. 6B–F), suggesting that HO-1 was required for EPA inhibition of macrophage M1 polarization. Additionally, silencing of *Hmox1* gene also abolished the protective effect of EPA on LPS-induced oxidative stress in RAW264.7 cells (Appendix Fig. S7).

## Macrophage HO-1 was required for EPA's protection against cardiomyocyte injury induced by M1 polarized macrophages

Given the important role of HO-1 in maintaining macrophage Mox phenotype and mediating EPA protection against M1 polarization, the following experiment further investigated whether macrophage HO-1 was required for EPA's protection against M1 polarized macrophage-induced cardiomyocyte injury, by co-culturing HL-1 cells with *Hmox1* gene-silenced, EPA-treated and LPS-challenged RAW264.7 cells (Fig. EV5A). As speculated, silencing *Hmox1* gene in RAW264.7 cells completely revoked the inhibitory effect of the EPA-treated RAW264.7 cells on HL-1 cells inflammation and oxidative stress (Fig. EV5B–J).

## EPA induced Mox polarization of macrophages in hearts of the diabetic mice

To verify the effect of EPA on Mox polarization of macrophages in vivo, *Hmox1* and *TrxR1* expression was determined in the diabetic heart tissue. Although mRNA levels of *Hmox1* and *TrxR1* were not affected by EPA, their proteins-decreased in the diabetic hearts-were upregulated by EPA (Fig. 7A–D). As these results were from cardiac tissues with cardiomyocyte being the predominant cell type, HO-1 and TrxR1 were further double-stained with the macrophage maker F4/80, respectively, showing enhanced colocations of HO-1 and TrxR1 with macrophages in the EPA-treated diabetic hearts (Fig. 7E–H).

## EPA promoted Mox polarization in human peripheral blood monocyte-derived macrophages from diabetic patients

In order to verify whether EPA could induce macrophage Mox polarization in diabetic patients, peripheral blood monocytes were collected from twelve patients with T2DM, induced into mature macrophages (Fig. 7I), and treated with EPA or the Mox inducer ox-PAPC. Both ox-PAPC and EPA was capable to increase protein expression of HO-1 and TrxR1 as hallmarks of Mox polarization (Fig. 7J–K). These results provided evidence for clinical application of EPA in the alteration of macrophage phenotype in diabetic patients.

The clinical information of subjects was provided in Appendix Table S1. The Mann–Whitney U test was used to predict whether there was a significant difference between the groups. The Mann–Whitney U test analysis showed that there was no significant difference in Age, HbA$_{1c}$, FPG, BMI and duration of T2DM between the groups ($n = 6$) ($P > 0.05$).

## EPA-activated HO-1 expression via GPR120 in M1 polarized macrophages

Subsequent investigations explored possible mechanism by which EPA induced HO-1 expression. Western blot analysis

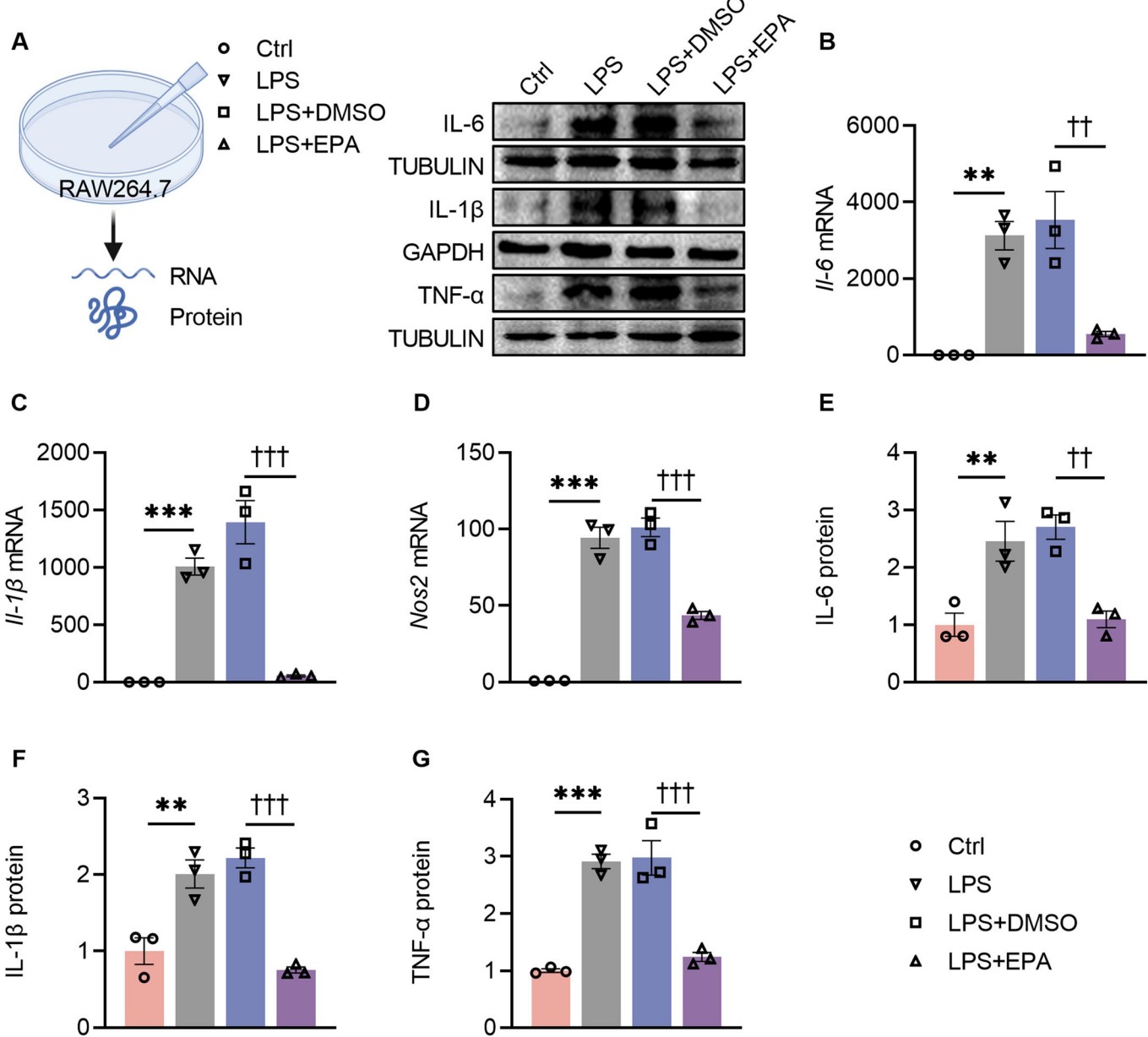

**Figure 4. EPA inhibited M1 polarization of macrophages.**

(A) Schematic representation of the experimental protocol. (B–D) mRNA expression of *Il-6*, *Il-1β*, and *Nos2* in RAW264.7 cells were detected by qRT-PCR. For (B), **$P = 0.001$, LPS vs. Ctrl; $^{††}P = 0.001$, LPS + EPA vs. LPS + DMSO. For (C, D), ***$P < 0.001$, LPS vs. Ctrl; $^{†††}P < 0.001$, LPS + EPA vs. LPS + DMSO. (E–G) Protein levels of IL-6, IL-1β, and TNF-α were determined using Western blot in RAW264.7 cells. For (E), **$P = 0.003$, LPS vs. Ctrl; $^{††}P = 0.001$, LPS + EPA vs. LPS + DMSO. For (F), **$P = 0.001$, LPS vs. Ctrl; $^{†††}P < 0.001$, LPS + EPA vs. LPS + DMSO. For (G), ***$P < 0.001$, LPS vs. Ctrl; $^{†††}P < 0.001$, LPS + EPA vs. LPS + DMSO. Data information: For (B–G), the data are normalized to Ctrl. Data are represented as individual data points of $n = 3$ (B–G) biological replicates and means ± SEM. Analysis by one-way ANOVA. The abbreviations are the same as in Figs. 1, 2 and 3. Groups: Ctrl control, LPS LPS-stimulated RAW264.7 cells, LPS + DMSO LPS-stimulated RAW264.7 cells treated DMSO, LPS + EPA LPS-stimulated RAW264.7 cells treated with EPA. Source data are available online for this figure.

showed that EPA increased the protein level of GPR120 (Fig. 8A), but not GPR40 (Appendix Fig. S8), indicating that GPR120, rather than GPR40, might be involved in EPA's elevation of HO-1. This speculation was verified using the GPR120 antagonist AH7614 (Fig. 8B), showing that EPA partially lost the capacity to increase HO-1 protein in the presence of AH7614 (Fig. 8C,D).

## Discussion

The present work reported that dietary supplementation with EPA prevented DC in a mouse model of T2DM. Mechanistically, EPA promoted Mox polarization of macrophages, the effect of which inhibited the M1 phenotype, attenuating cardiomyocyte injury. Moreover, HO-1 was identified to be a keystone to maintain the

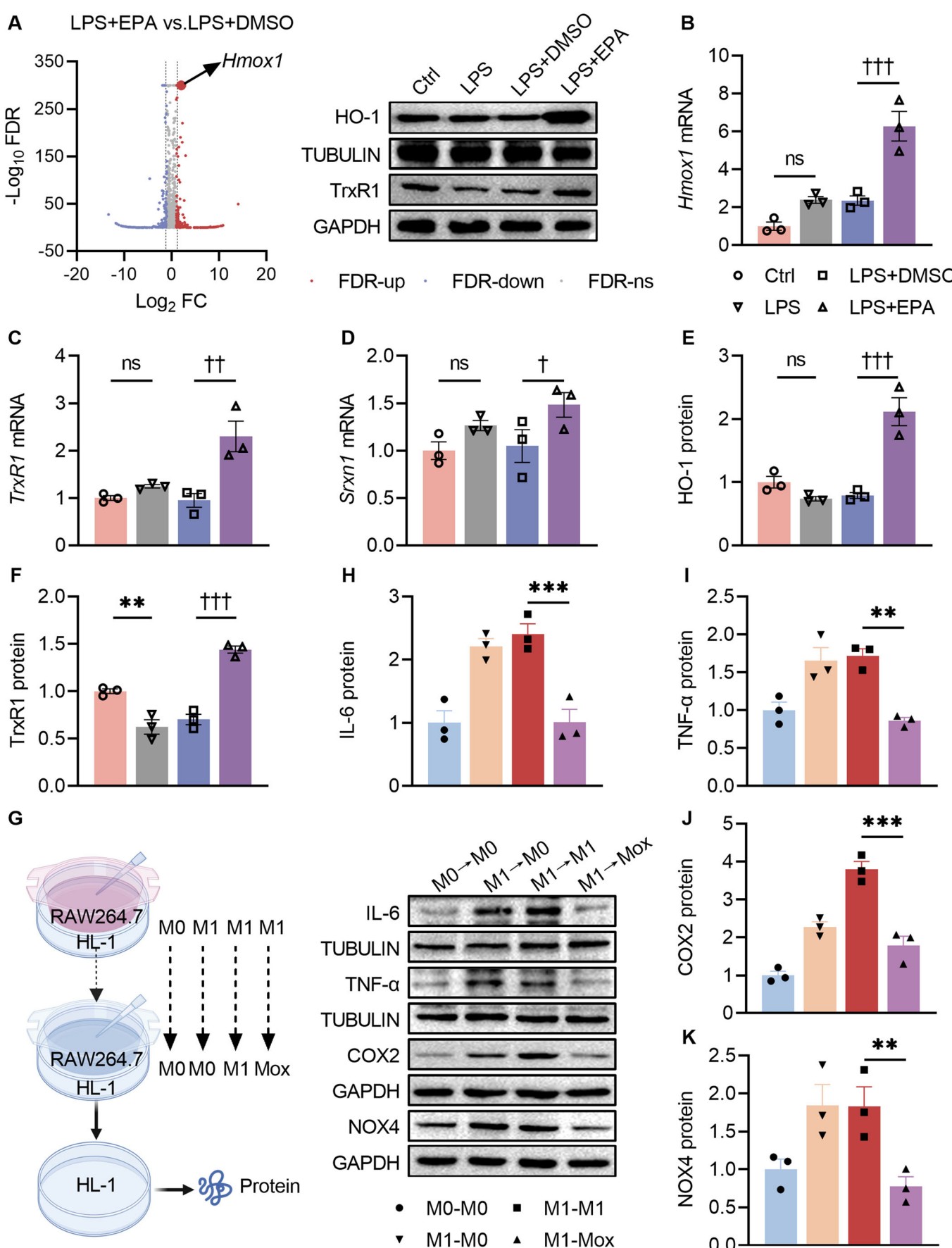

**Figure 5. EPA promoted Mox polarization of macrophages, and Mox polarized macrophages combated M1 polarization-induced cardiomyocyte injury.**

(A) Volcano plot of RNA-seq showed a significant upregulation of *Hmox1* mRNA by EPA compared with DMSO in LPS-stimulated RAW264.7 cells. (B–D) RAW264.7 cell mRNA expression of *Hmox1*, *TrxR1*, and *Srxn1* were determined by qRT-PCR. For (B), $^{†††}P < 0.001$, LPS + EPA vs. LPS + DMSO. For (C), $^{††}P = 0.001$, LPS + EPA vs. LPS + DMSO. For (D), $^{†}P = 0.035$, LPS + EPA vs. LPS + DMSO. (E, F) Protein expression of HO-1 and TrxR1 were determined by Western blot. For (E), $^{†††}P < 0.001$, LPS + EPA vs. LPS + DMSO. For (F), $^{**}P = 0.001$, LPS vs. Ctrl; $^{†††}P < 0.001$, LPS + EPA vs. LPS + DMSO. (G) Schematic representation of the experimental protocol. The arrows represent shifts to RAW264.7 cells phenotypes. Briefly, HL-1 cells were divided into 4 groups, and co-cultured with M0, M1, M1, and M1 phenotypes of RAW264.7 cells respectively for 24 h, followed by substitutions with novel M0, M0, M1 and Mox phenotypes of RAW264.7 cells respectively for another 24 h. (H–K) Protein expression of IL-6, TNF-α, COX2, and NOX4 were determined by Western blot in HL-1 cells. For (H), $^{***}P < 0.001$, M1→Mox vs. M1 → M1. For (I), $^{**}P = 0.001$, M1→Mox vs. M1 → M1. For (J), $^{***}P < 0.001$, M1→Mox vs. M1 → M1. For (K), $^{**}P = 0.007$, M1→Mox vs. M1 → M1. Data information: For (B–F), the data are normalized to Ctrl. For (H–K), the data are normalized to M0 → M0. Data are represented as individual data points of $n = 3$ (B–F, H–K) biological replicates and means ± SEM. ns, not significant. Analysis by one-way ANOVA. *Hmox1*/HO-1 heme oxygenase 1, *Srxn1* sulfiredoxin 1, *TrxR1*/TrxR1 thioredoxin reductase 1. Other abbreviations are the same as in Figs. 1, 2 and 3. Groups: Ctrl control, LPS LPS-stimulated RAW264.7 cells, LPS + DMSO LPS-stimulated RAW264.7 cells treated with DMSO, LPS + EPA LPS-stimulated RAW264.7 cells treated with EPA. M0, RAW264.7 cells in a resting state without polarization; M1, RAW264.7 cells challenged with LPS; Mox, RAW264.7 cells treated with ox-PAPC to induced an antioxidant phenotype. Source data are available online for this figure.

Mox phenotype and mediating EPA's beneficial effects. GPR120 might be involved in EPA's elevation of HO-1. The capability of EPA to induce macrophage Mox polarization was further confirmed in peripheral blood monocytes-derived macrophages from patients with T2DM. To date, this has been the first study reporting the effect of EPA on DC, and the role of Mox polarized macrophage in cardiomyocyte injury.

The current strategies for DC intervention are still far from satisfactory. The elucidation of molecular mechanism and development of novel effective approaches are essential for DC management. Under diabetic condition, cardiac inflammation and oxidative stress boost mutually, forming a vicious circle that results in detrimental effects, such as cardiac apoptosis, fibrosis, as well as ventricular structure remodeling and dysfunction, increasing the risk of heart failure in diabetic patients (Dillmann, 2019; Lee et al, 2012; Ritchie and Abel, 2020). Therefore, successful inhibition of inflammation or oxidative stress is a viable strategy that may lead to the breakdown of this DC-accelerating circle.

The immune system provides a protective inflammatory response necessary for host defense from infections or invasions. The majority of leukocytes present in the resting heart are F4/80$^+$ macrophages in close proximity to cardiomyocytes (Hulsmans et al, 2017; Hulsmans et al, 2016; Munshi, 2017). M1 polarized macrophages are major producers of inflammatory mediators during inflammatory diseases including DC, contributing to cardiac dysfunction (Gordon and Martinez-Pomares, 2017; Jia et al, 2017; Qiu et al, 2019; Sreedhar et al, 2017). In line with these findings, the dominance of M1 polarized macrophages in the diabetic hearts was found in the present work (Fig. 2I–L). Hence, a shift in macrophage phenotypes-from M1 to other protective phenotypes-should yield beneficial outcomes in DC.

Mox is an antioxidant phenotype that is induced by oxidized phospholipids, possessing an antioxidant capacity (Kadl et al, 2010). This antioxidant effect may result from the expression of nuclear factor erythroid2-related factor 2 (NRF2)-dependent antioxidant genes, including *Hmox1*, *Srxn1*, *TrxR1*, glutathione reductase and synthase, etc (Kadl et al, 2010; Serbulea et al, 2018a). In adipose tissue of obese mice, enhanced M1 polarization and repressed Mox polarization were found in macrophages, along with an impairment of antioxidant capacity (Zhang et al, 2022). In the present study, Mox polarization degraded the M1 phenotype of macrophages, and diminished M1 polarization-induced

cardiomyocyte injury (Fig. 5G–K), providing compelling evidence for activation of Mox polarization in future management of DC. One advantage of this study would be the verification of EPA's effect on the M1-Mox shift in peripheral blood monocytes-derived macrophages from patients with T2DM (Fig. 7J,K). This finding, together with the animal and cell studies, would promote bench-to-bedside translation of EPA supplementation in DC.

In spite of the beneficial effects of Mox polarization found in our study as well as others' work (Kadl et al, 2010; Serbulea et al, 2018a; Serbulea et al, 2018b), a few studies reported a detrimental role of Mox polarization in atherosclerosis (Seimon and Tabas, 2009; Skuratovskaia et al, 2020). The Mox phenotype was found to have weaker phagocytosis and migration capacities compared with those of the M1 and M2 phenotypes (Chistiakov et al, 2015; Leitinger and Schulman, 2013). This hampers the removal of excessive lipids in the arterial wall. However, it is also believed that the weaker phagocytosis ability and powerful antioxidant capacity of Mox macrophages protected themselves from becoming foam cells, which prevented the formation of the atherosclerotic plaque (Li et al, 2022). Despite the controversy in delineating the role of Mox polarization in atherosclerosis, one possible explanation for its protective effect on DC could be explained by the pathophysiology of the disease. Atherosclerosis is characterized by lipid accumulation. Thus, the phagocytosis capacity of macrophages during atherogenesis is crucial. However, the inflammatory status of macrophages might influence more on the pathogenesis of DC, as compared with their phagocytosis activity. It is therefore not surprising that the Mox phenotype of macrophages played a protective role in DC in the present study.

One interesting finding of the current research is that HO-1 determined the Mox phenotype of macrophages. HO-1, TrxR1, SRXN1, and many other antioxidants are downstream targets of NRF2 (Kadl et al, 2010). Despite the acknowledgement of HO-1, TrxR1, and SRXN1 as hallmarks of Mox polarization, the roles of these antioxidants in maintaining the Mox phenotype were not previously known. Our study provided the first-hand evidence that HO-1 was a keystone in this phenotype, since knockdown of HO-1 expression blunted TrxR1-another hallmark of Mox-expression (Appendix Fig. S6D). This finding also suggested that HO-1 might positively regulate TrxR1 expression, through either direct or indirect mechanisms. Therefore, HO-1 played a central role in Mox polarization. However, the mechanism by which HO-1 regulates

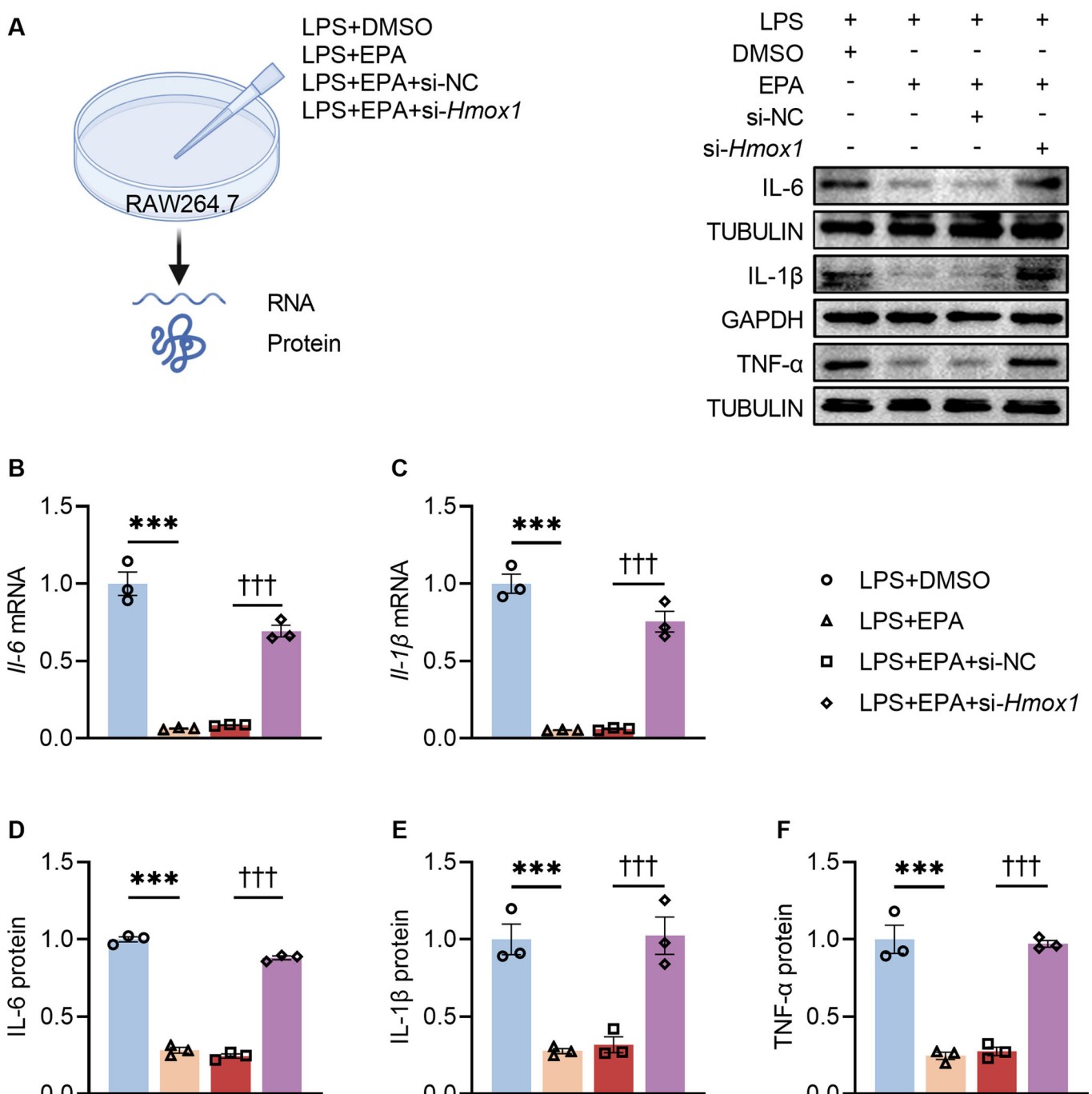

**Figure 6. HO-1 played a critical role in EPA suppression of macrophage M1 polarization.**

(A) Schematic representation of the experimental protocol. (B, C) mRNA expression of *Il-6* and *Il-1β* in RAW264.7 cells. ***P < 0.001, LPS + EPA vs. LPS + DMSO; †††P < 0.001, LPS + EPA+ si-*Hmox1* vs. LPS + EPA+si-NC. (D–F) Protein levels of IL-6, IL-1β, and TNF-α. ***P < 0.001, LPS + EPA vs. LPS + DMSO; †††P < 0.001, LPS + EPA+ si-*Hmox1* vs. LPS + EPA+si-NC. Data information: For (B–F), the data are normalized to LPS + DMSO. Data are represented as individual data points of n = 3 (B–F) biological replicates and means ± SEM. Analysis by one-way ANOVA. si-*Hmox1 Hmox1* siRNA, si-NC negative control siRNA. Other abbreviations are the same as in Figs. 1, 2, 3 and 5. Groups: LPS + DMSO LPS-stimulated RAW264.7 cells treated with DMSO, LPS + EPA LPS-stimulated RAW264.7 cells treated with EPA, LPS + EPA+si-NC LPS-stimulated RAW264.7 cells treated with EPA and si-NC, LPS + EPA+si-*Hmox1* LPS-stimulated RAW264.7 cells treated with EPA and si-*Hmox1*. Source data are available online for this figure.

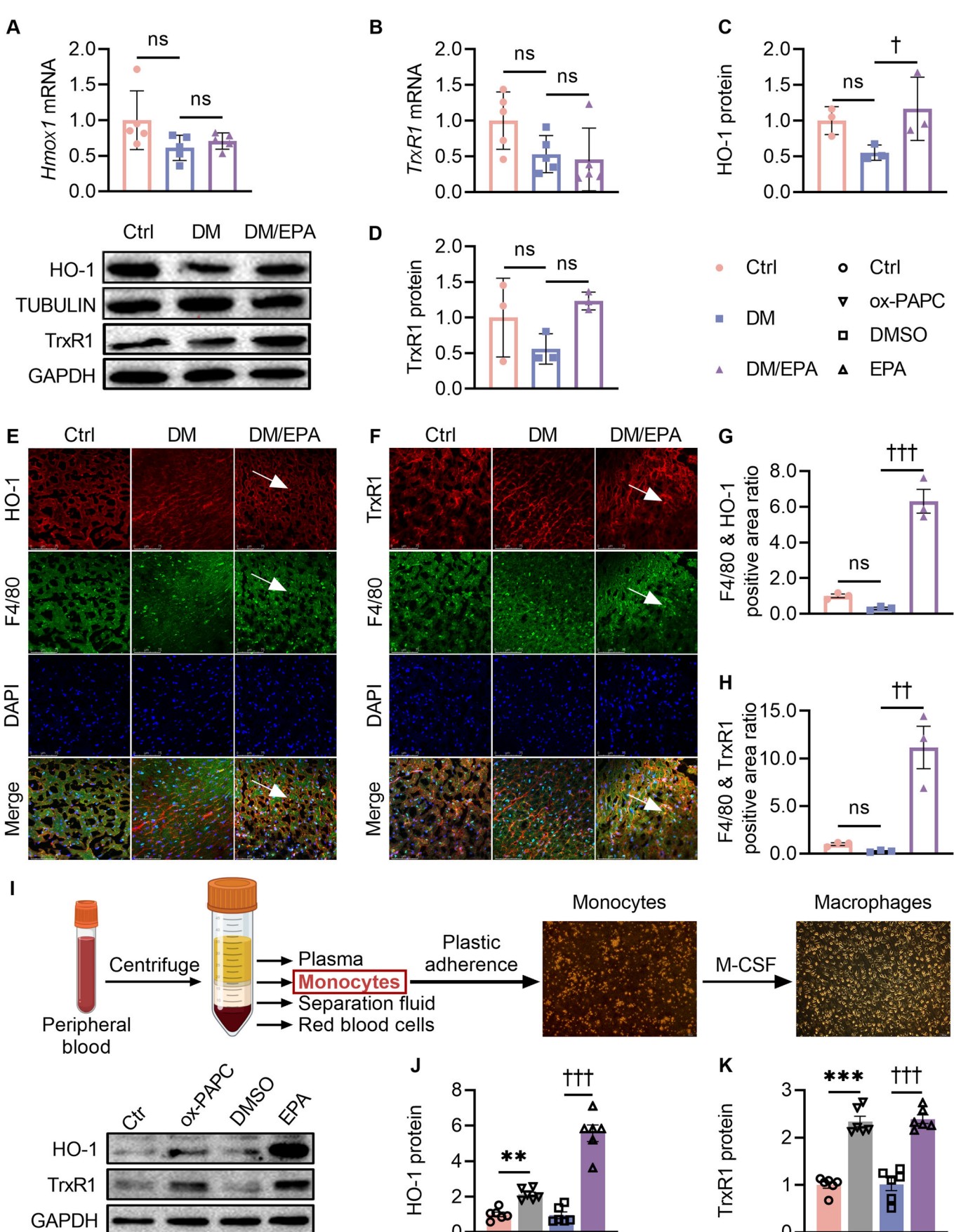

**Figure 7. EPA promoted Mox polarization of macrophages in diabetic mice and T2DM patients.**

(A, B) Cardiac mRNA expression of *Hmox1* and *TrxR1* in the diabetic mice. (C, D) Cardiac protein expression of HO-1 and TrxR1 in the diabetic mice. For (C), [†]$P = 0.039$, DM/EPA vs. DM. (E) Immunofluorescence of HO-1 (red) and F4/80 (green) (bar = 75 μm). The white arrows point to regions with HO-1 and F4/80 fluorescence. (F) Immunofluorescence of TrxR1 (red) and F4/80 (green) (bar = 75 μm). The white arrows point to regions with TrxR1 and F4/80 fluorescence. (G, H) Quantification of merged immunofluorescence of F4/80 with either HO-1 or TrxR1. For (G), [†††]$P < 0.001$, DM/EPA vs. DM. For (H), [††]$P = 0.001$, DM/EPA vs. DM. (I) Schematic diagram for isolation and differentiation of human peripheral blood monocytes (bar=100 μm). (J, K) Protein levels of HO-1 and TrxR1 in human peripheral blood monocyte-derived macrophages. For (J), [**]$P = 0.007$, ox-PAPC vs. Ctrl; [†††]$P < 0.001$, EPA vs. DMSO. For (K), [***]$P < 0.001$, ox-PAPC vs. Ctrl; [†††]$P < 0.001$, EPA vs. DMSO. Data information: For (A–D, G, H, J, K), the data are normalized to Ctrl. Data are represented as individual data points of $n = 5, 5, 5$ (A, B), $n = 3, 3, 3$ (C, D), $n = 3, 3, 3$ (G, H), $n = 6, 6, 6$ (J, K) biological replicates and means ± SEM. ns, not significant. Analysis by one-way ANOVA. M-CSF macrophage colony-stimulating factor, ox-PAPC oxidized 1-palmitoyl-2-arachidonoyl-sn-glycero-3-phosphocholine. Other abbreviations are the same as in Figs. 1, 2 and 5. Groups: For animal studies, Ctrl control, DM diabetes mellitus, DM/EPA diabetic mice supplemented with EPA. For human studies, Ctrl control, ox-PAPC ox-PAPC-stimulated human macrophages, DMSO DMSO-stimulated human macrophages, EPA EPA-stimulated human macrophages. Source data are available online for this figure.

TrxR1 expression remains unclear. In addition, the roles of TrxR1, SRXN1 and even other antioxidants in maintaining Mox phenotype remains unknown. These questions need to be studied in the future.

Another innovation of the present study is the identification of EPA as an effective approach for DC intervention. Although EPA has been studied under various health and disease conditions (Zhang et al, 2019), its effect on DC was not previously known. EPA was reported to trigger M2 polarization in obese adipose tissue and hypertensive cardiomyopathy (Gharraee et al, 2022; Tian et al, 2020). However, in the present work, EPA did not activate M2 polarization, but switched M1 to Mox phenotype, alleviating the M1 polarization-induced cardiomyocyte injury. To date, this has been the first report on the EPA-induced M1 to Mox transition. The distinct effects of EPA on macrophage polarization might be explained by the difference in diseases, which could cause differential cellular response to EPA. This notion is supported by the finding that EPA improved post-myocardial infarction cardiac remodeling in mice by inhibiting M1, without activating M2 polarization (Takamura et al, 2017).

This study found that EPA induced HO-1 expression via GPR120. However, the mechanism by which HO-1 expression was activated following the EPA-GPR120 binding remains unknown. NRF2 is a canonical transcription factor for *Hmox1*. EPA might possibly activate *Hmox1* expression through NRF2. Kelch-like ECH associated protein 1 (KEAP1) is a negative regulator of NRF2, sequestering NRF2 in the cytoplasm and preventing its nuclear translocation (Suzuki and Yamamoto, 2015). In response to exogenous stimuli, NRF2 disassociates from KEAP1 and binds to antioxidant response elements (ARE), and then activates the transcription of downstream antioxidant genes including *Hmox1* (Nguyen et al, 2003; Tkachev et al, 2011). It was reported that KEAP1 could be bound by transforming growth factor-β-activated kinase 1 (TAK1), and inhibition of TAK1 by the compound 4F5C-QAME weakened the interaction between TAK1 and KEAP1, inhibiting NRF2 ubiquitination and the following degradation (Ge et al, 2023). This led to NRF2-induced *Hmox1* gene expression (Ge et al, 2023). Interestingly, DHA, another n-3 PUFA, was shown to bind GPR120, resulting in the inhibition of TAK1 (Oh and Olefsky, 2016). We therefore speculated EPA might increase HO-1 level through GPR120/TAK1/KEAP1/NRF2 pathway. Additionally, in gingival epithelial cells, GPR120 was indicated to be an upstream regulator of the ERK/NRF2/ARE pathway, suggesting that EPA might also activate *Hmox1* expression via GPR120/ERK/NRF2 (Yokoji-Takeuchi et al, 2020). These potential pathways, although putative, might at least provide clues for the EPA-elevated HO-1 level.

In the present work, EPA decreased blood glucose levels in the diabetic mice starting from the 20th week post DM onset (Fig. EV1B). It would be suspicious that EPA might prevent DC through its improvement of hyperglycemia, but not macrophage polarization. However, EPA had already protected against cardiac dysfunction 12 weeks post DM (Fig. 1A–D). This finding demonstrated that EPA prevented DC at least in part independent from its hypoglycemic effect.

DHA is another important n-3 PUFA that can be transformed from EPA under C20-elongase (Niu et al, 2009). To date, there has been only one study reporting that DHA protected against palmitate-induced mitochondrial dysfunction in DC (Gui et al, 2020). Apart from novelty, another reason to study EPA other than DHA was that EPA was reported to be more effective than DHA in improving hyperglycemia and insulin resistance in db/db mice (Zhuang et al, 2021), and in balancing pro-inflammatory and anti-inflammatory cytokines in a clinical trial of individuals with chronic inflammation (So et al, 2021). Nevertheless, it would be interesting to compare the effects and molecular mechanisms of EPA and DHA, even fish oil and krill oil as good sources of n-3 PUFAs, on DC in future studies.

Taken together, the present study found a profound protective effect of EPA on DC through shifting macrophage M1 to Mox polarization. This work may provide EPA as an effective approach, and Mox polarization as a viable strategy, for DC intervention.

## Methods

### Reagents and tools table

| Reagent/Resource | Reference or Source | Identifier or Catalog Number |
|---|---|---|
| **Experimental Models** | | |
| Human peripheral blood | Qilu Hospital of Shandong University | N/A |
| C57BL/6N (*M. musculus*) | Charles River Laboratories | N/A |
| HL-1 cell line | Procell | Cat# CL-0605 |
| RAW264.7 cell line | Procell | Cat# CL-0190 |
| **Antibodies** | | |
| Mouse monoclonal anti-8-OHdG IHC (1:200) | Santa Cruz Biotechnology | Cat# sc-393871 RRID: AB_2892631 |

| Reagent/Resource | Reference or Source | Identifier or Catalog Number |
|---|---|---|
| Rabbit polyclonal anti-ARG1 WB (1:1000) | ABclonal | Cat# A1847 RRID: AB_2763883 |
| Mouse monoclonal anti-CD206 IF (1:100) | Santa Cruz Biotechnology | Cat# sc-58986, RRID: AB_2144945 |
| Mouse monoclonal anti-CD86 IF (1:100) | Santa Cruz Biotechnology | Cat# sc-28347 RRID: AB_627200 |
| Rabbit monoclonal anti-COX2 WB (1:1000) | Cell Signaling Technology | Cat# 12282 RRID: AB_2571729 |
| Rabbit polyclonal anti-F4/80 IF (1:100), IHC (1:500) | Proteintech | Cat# 29414-1-AP RRID: AB_2918300 |
| Mouse monoclonal anti-GAPDH WB (1:20000) | Proteintech | Cat# 60004-1-Ig RRID: AB_2107436 |
| Rabbit polyclonal anti-GPR40 WB (1:1000) | Affinity Biosciences | Cat# DF8146, RRID:AB_2841477 |
| Rabbit polyclonal anti-GPR120 WB (1:1000) | Affinity Biosciences | Cat# AF5219 RRID: AB_2837705 |
| Mouse monoclonal anti-HO-1 IF (1:100), WB (1:1000) | Proteintech | Cat# 66743-1-Ig RRID: AB_2882091 |
| Mouse monoclonal Anti-iNOS IF (1:100) | Santa Cruz Biotechnology | Cat# sc-7271 RRID: AB_627810 |
| Mouse monoclonal anti-IL-1β IHC (1:200), WB (1:1000) | Cell Signaling Technology | Cat# 12242 RRID: AB_2715503 |
| Mouse monoclonal anti-IL-6 WB (1:1000) | Proteintech | Cat# 66146-1-Ig RRID: AB_2881543 |
| Mouse monoclonal anti-IL-10 WB (1:1000) | Santa Cruz Biotechnology | Cat# sc-365858 RRID: AB_10859554 |
| Mouse monoclonal anti-NOX2 WB (1:1000) | Santa Cruz Biotechnology | Cat# sc-130543 RRID: AB_2261483 |
| Rabbit polyclonal anti-NOX4 WB (1:1000) | Proteintech | Cat# 14347-1-AP RRID: AB_10638146 |
| Rabbit polyclonal anti-TGF-β1 WB (1:1000) | Cell Signaling Technology | Cat# 3711 RRID: AB_2063354 |
| Rabbit polyclonal anti-TNF-α WB (1:1000) | ABclonal | Cat# A11534 RRID: AB_2758597 |
| Rabbit monoclonal anti-TNF-α WB (1:1000) | Cell Signaling Technology | Cat# 11948 RRID: AB_2687962 |
| Mouse monoclonal anti-TrxR1 IF (1:200), WB (1:1000) | Santa Cruz Biotechnology | Cat# sc-28321 RRID: AB_628405 |
| Rabbit polyclonal anti-TUBULIN WB (1:1000) | Proteintech | Cat# 10068-1-Ap RRID: AB_2303998 |
| Alexa Fluor® 488-conjugated Affinipure Goat Anti-Rabbit IgG (1:500) | Jackson ImmunoResearch Labs | Cat# 111-545-003 RRID: AB_2338046 |
| Alexa Fluor® 594-conjugated Affinipure Goat Anti-Mouse IgG (1:500) | Jackson ImmunoResearch Labs | Cat# 115-585-003 RRID: AB_2338871 |
| Peroxidase-conjugated Affinipure Goat Anti-Rabbit IgG (1:20,000) | Jackson ImmunoResearch Labs | Cat# 111-035-003 RRID: AB_2313567 |
| HRP-conjugated Affinipure Goat Anti-Mouse IgG (1:5000) | Proteintech | Cat# SA00001-1 RRID: AB_2722565 |
| **Oligonucleotides and other sequence-based reagents** | | |
| PCR primers | This study | Table S3 |

| Reagent/Resource | Reference or Source | Identifier or Catalog Number |
|---|---|---|
| **Chemicals, Enzymes and other reagents** | | |
| 4′,6-diamidino-2-phenylindole (DAPI) staining solution | Beyotime | Cat# C1005 |
| AH7614 | MCE | Cat# HY-19996 |
| Interleukin-4 (IL-4) | MCE | Cat# HY-P7080 |
| Interleukin-13 (IL-13) | MCE | Cat# HY-P70460 |
| Lipopolysaccharides (LPS) | Solarbio | Cat# L8880 |
| Macrophage colony-stimulating factor (M-CSF) | MCE | Cat# HY-P701101 |
| Eicosapentaenoic acid (EPA) (for cell) | MCE | Cat# HY-B0660 |
| Eicosapentaenoic acid (EPA) (for animal) | Aladdin | Cat# E135694 |
| Dihydroethidium (DHE) | Beyotime | Cat# S0063 |
| Streptozotocin (STZ) | Sigma-Aldrich | Cat# S0130 |
| oxidized 1-palmitoyl-2-arachidonoyl-sn-glycero-3-phosphocholine (ox-PAPC) | Sigma-Aldrich | Cat# 870604P |
| Advanced DNA RNA Transfection Reagen | ZETA LIFE | Cat# AD600025 |
| LightCycler 480 SYBR Green I Master | Roche LifeScience | Cat# 04707516001 |
| **Software** | | |
| Image-Pro Plus software | Media Cybernetics | http://www.licor.com/bio/image-studio-lite/ |
| SPSS 26.0 | IBM | https://www.ibm.com/software |
| GraphPad Prism 9.0 | GraphPad | http://www.graphpad.com/ |
| **Other** | | |
| Blood Glucose Content Test Kit | Solarbio | Cat# BC2495 |
| E.Z.N.A.®Total RNA Kit I | Omega BIO-TEK | Cat# R6834-02 |
| PrimeScript RT reagent Kit | Takara Biomedical Technology | Cat# RR037B |
| One Step TUNEL Apoptosis Assay Kit | Beyotime | Cat# C1090 |
| Modified Masson's Trichrome Stain Kit | Solarbio | Cat# G1346 |
| Human peripheral blood mononuclear cell isolation kit | Tbdscience | Cat# TBD2011H05 |
| WGA Lectin (FITC) Assay Kit | GeneTex | Cat# GTX01502 |
| Enhanced BCA Protein Assay Kit | Beyotime | Cat# P0010 |
| Sensitive ECL Chemiluminescent Detection Kit | Proteintech | Cat# PK10002 |

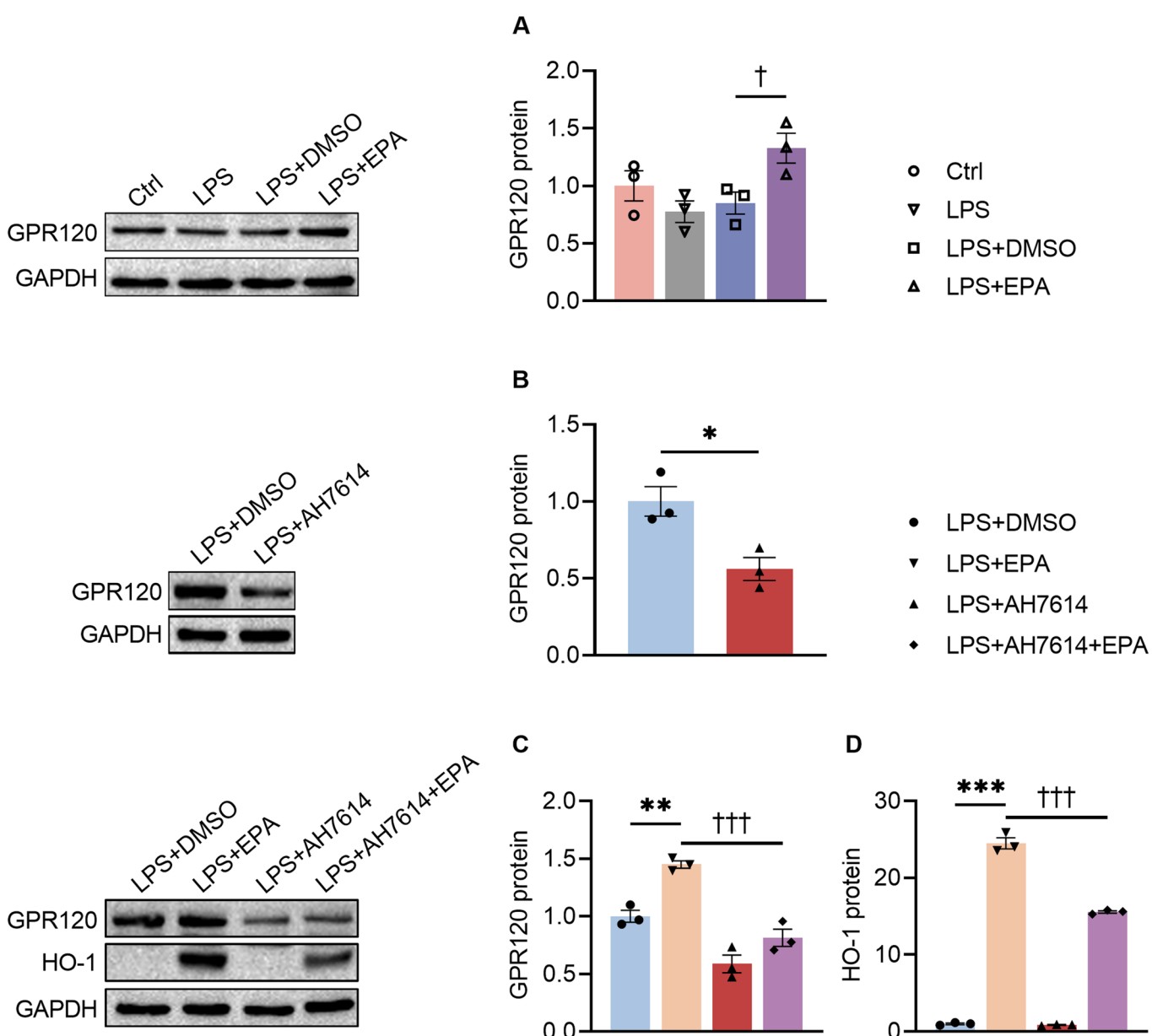

**Figure 8. EPA activated HO-1 expression via GPR120 in M1 polarized macrophages.**

(A) Protein levels of GPR120. $^{†}P = 0.018$, LPS + EPA vs. LPS + DMSO. Analysis by one-way ANOVA. (B) To verify the efficacy of AH7614, protein level of GPR120 was determined in RAW264.7 cells. $^{*}P = 0.023$, LPS + AH7614 vs. LPS + DMSO. Analysis by two-tailed unpaired Student's t-test. (C, D) To study the effect of GPR120 in the maintenance of EPA's elevation of HO-1, protein levels of GPR120 and HO-1 were measured in RAW264.7 cells. For (C), $^{**}P = 0.001$, LPS + EPA vs. LPS + DMSO; $^{†††}P < 0.001$, LPS + AH7614 + EPA vs. LPS + EPA. For (D), $^{***}P < 0.001$, LPS + EPA vs. LPS + DMSO; $^{†††}P < 0.001$, LPS + AH7614 + EPA vs. LPS + EPA. Analysis by one-way ANOVA. Data information: For (A), the data are normalized to Ctrl. For (B–D), the data are normalized to LPS + DMSO. Data are represented as individual data points of $n = 3$ (A–D) biological replicates and means ± SEM. GPR120 G-protein-coupled receptor 120, AH7614 4-methylN-9H-xanthen-9-yl-benzenesulfonamide, a potent and selective GPR120 antagonist. Other abbreviations are the same as in Figs. 1, 3 and 5. Groups: Ctrl control, LPS LPS-stimulated RAW264.7 cells, LPS + DMSO LPS-stimulated RAW264.7 cells treated with DMSO, LPS + EPA LPS-stimulated RAW264.7 cells treated with EPA, LPS + AH7614 LPS-stimulated RAW264.7 cells treated with AH7614, LPS + AH7614 + EPA LPS-stimulated RAW264.7 cells treated with AH7614 and EPA. Source data are available online for this figure.

## In vivo animal studies

C57BL/6N male mice were obtained from Charles River Laboratories (Beijing, CHN), and were housed in the Animal Center of Shandong University at 22–24 °C, in a 12 h/12 h light/dark cycle with adequate water and food supplies. The experimental procedures were approved by the Ethics Committee of Preventive Medicine of Shandong University (Permission number: SYKX20200022). After adaptation, the 8-week-old mice were randomly divided into the nondiabetic control group (Ctrl), diabetic model group (DM) and EPA-treated diabetic group (DM/EPA). To establish a model of T2DM, the mice were

intraperitoneally injected with STZ (Sigma-Aldrich, Shanghai, CHN; dissolved in 0.1 mg/l sodium citrate, pH 4.5), at a dose of 50 mg/kg·body weight per day, for 5 consecutive days. The Ctrl mice received equal volumes of sodium citrate. One week after the last injection of STZ, fasting blood glucose levels (6-h fast) were measured, with a value above 13.89 mmol/l considered as DM. Immediately after the confirmation of DM, the Ctrl, DM and DM/EPA groups were fed a standard AIN-93G diet, HFD or a 2% EPA (Aladdin, Shanghai, CHN)-containing HFD (the composition of the experimental diets were listed in Appendix Table S2), respectively, for a total period of 24 weeks. During the experimental procedure, body weight, FBG, glucose tolerance, insulin tolerance and cardiac function were monitored.

At the end of the experimental procedure, the mice were euthanized under anesthesia with isoflurane (RWD, Shenzhen, Guangdong, CHN), with heart tissues harvested for analysis.

## Human studies

### Study subjects

Twelve T2DM patients from Qilu Hospital of Shandong University were recruited in the study. The clinical information of subjects was presented in Table 1. This research was approved by the Public Health Ethics Committee of Shandong University (Ethics Approval No. LL20230101). The study conformed to the principles set out in the WMA Declaration of Helsinki and the Department of Health and Human Services Belmont Report.

### Inclusion criteria

Subjects are eligible to be included in the study only if all of the following criteria apply:

1. Informed consent obtained before study.
2. Male or female, age between 30 and 65 years at the time of signing informed consent.
3. Diagnosis of T2DM.

### Exclusion criteria

Subjects are excluded from the study if any of the following criteria apply:

1. Diagnosis of liver or renal function insufficiency, hematologic disorders or autoimmune diseases.
2. Participation in clinical trial within 90 days.
3. Treatment with antibiotics, corticosteroid within 90 days.
4. Presence of acute pancreatitis within the past 180 days.
5. History or presence of chronic pancreatitis.
6. History of malignant neoplasms within the past 5 years.
7. History of smoking, known or suspected abuse of alcohol or recreational drugs.

### Peripheral blood monocyte isolation, culture, and treatment

Human peripheral blood monocytes were isolated using human monocyte isolation kit (Tbdscience, Tianjin, CHN) with density gradient centrifugation according to the manufacturer's instructions. Monocytes were further purified by plastic adherence as described before (Delirezh et al, 2013). After 2 h of cell adhesion, non-adherent cells were then aspirated and removed. The remaining adherent monocytes [in RPMI1640 medium (Thermo Fisher Scientific, Shanghai, CHN) containing 10% FBS] were washed with PBS and matured into macrophages using 50 ng/ml macrophage colony-stimulating factor (M-CSF, MCE), for a period of 7 days. At day 3, half of the culture medium was changed, and supplemented with M-CSF, forming a concentration of 50 ng/ml. To study the effect of EPA on Mox polarization, the macrophages were treated with ox-PAPC (50 μg/ml), EPA (50 μg/ml) or the vehicle DMSO for 24 h. The cells were then harvested, with Mox polarization assessed using Western blot.

### Cell culture and experiments

HL-1, a mouse cardiomyocyte cell line (mycoplasma free cells), verified by STR profiling test, was purchased from Procell (Wuhan, Hubei, CHN), culturing in *DMEM* (4.5 g/l glucose; Procell) containing 10% FBS. The mouse macrophage cell line RAW264.7 (mycoplasma free cells), verified by STR profiling test, was provided by Procell, culturing in a RAW264.7 cell-specific medium (Procell) contains *DMEM* (4.5 g/l glucose) and 10% FBS.

M1 polarization was induced in RAW264.7 cells using lipopolysaccharides (LPS, 100 ng/ml, Solarbio, Beijing, CHN) for 6 h. A model of M1-polarized macrophage-induced cardiomyocyte injury was established by co-culture of HL-1 cells with LPS-stimulated RAW264.7 cells for 24 h, using transwell chambers.

To study the effect of EPA on M1 polarization of RAW264.7 cells, RAW264.7 cells were pretreated with EPA (50 μg/ml, MCE, Shanghai, CHN) or the vehicle DMSO for 18 h, followed by a 6-h LPS (100 ng/ml) stimulation. RNA-seq was performed to screen key molecular pathways or targets of EPA, as described in our previous work (Fan et al, 2023).

To investigate the effect of EPA-modulated macrophage M1 polarization on cardiomyocyte injury, EPA- or DMSO-pretreated M1 polarized RAW264.7 cells were incubated with HL-1 cells for 24 h, followed by assessment of RNA and protein expression using RNA-seq, qRT-PCR and Western blot. To investigate the effect of Mox polarized macrophages on cardiomyocyte injury, Mox polarized RAW264.7 cells induced by ox-PAPC (50 μg/ml, 24 h treatment, Sigma-Aldrich) were co-cultured with HL-1 cells for 24 h. To test whether HO-1 maintained the state of Mox polarization of macrophages, the Mox polarized RAW264.7 cells were treated with a *Hmox1* siRNA (sequence: CAACAGUGGCA-GUGGGAAUTT) or a negative control siRNA, for 24 h to collect RNA and 48 h for protein. In order to verify the role of macrophage HO-1 in mediating EPA's protection against M1-polarized macrophage-induced cardiomyocyte injury, *Hmox1* gene was silenced using a small interfering RNA in M1-polarized RAW264.7 cells in the presence of EPA (50 μg/ml), and co-cultured with HL-1 cells for 24 h.

For siRNA experiments, siRNA (20 μM) were transfected using the Advanced DNA RNA Transfection Reagent™ (ZETA LIFE, Menlo Park, California, USA) according to the manufacturer's protocol. Briefly, RAW264.7 cells were planted on the cell culture plate one day in advance, the cell confluence degree was up to 30–50% at the time of transfection, siRNA was directly mixed with transfection reagent according to 1:1 relationship, then mixed by blowing into a pipette for 10–15 times. Following incubation at room temperature for 10–15 min, the complex was added to the cell culture plates, mix gently, and incubated for 24 h. All the siRNAs

**Table 1. Clinical information of subjects.**

| Characteristic | $n = 12$ |
|---|---|
| Age (year) | 54.83 ± 2.77 |
| Male | $n = 6$ |
| Female | $n = 6$ |
| HbA$_{1c}$ (%) | 9.48 ± 0.46 |
| FPG (mmol/l) | 11.61 ± 0.93 |
| BMI (kg/m²) | 24.44 ± 0.79 |
| Duration of T2DM (year) | 10.42 ± 2.09 |

*BMI* body mass index, *FPG* fasting plasma glucose, *HbA$_{1c}$* glycosylated hemoglobin.

were designed by Generalbiol (Chuzhou, Anhui, CHN), and transfection efficiency was measured using qRT-PCR and Western blot.

M2 polarization was induced in RAW264.7 cells using interleukin-4 (IL-4, 20 ng/ml, MCE) and interleukin-13 (IL-13, 20 ng/ml, MCE) for 24 h. To study the effect of EPA on M2 polarization of RAW264.7 cells, RAW264.7 cells were treated with IL-4, IL-13, and EPA (50 μg/ml) or the vehicle DMSO for 24 h.

To study the effect of EPA on GPR120 and GPR40 of M1 polarized macrophages, RAW264.7 cells were pretreated with EPA (50 μg/ml) or the vehicle DMSO for 18 h, followed by a 6-h LPS (100 ng/ml) stimulation.

To examine the inhibitory effect of the GPR120 antagonist AH7614 (4-methylN-9H-xanthen-9-yl-benzenesulfonamide, 100 μM, MCE) on GPR120 in M1 polarized macrophages, RAW264.7 cells were pretreated with AH7614 (100 μM) or the vehicle DMSO for 1 h, followed by a 6-h LPS (100 ng/ml) stimulation in the presence of AH7614. To investigate whether GPR120 mediates the EPA's elevation of HO-1, macrophages were pretreated with AH7614 (100 μM) for 1 h, and then stimulated with LPS (100 ng/ml) with or without EPA (50 μg/ml) for 6 h in the presence of AH7614. RAW264.7 cells were stimulated LPS and DMSO for 6 h as the solvent control.

### Glucose tolerance test (GTT)

GTT was carried out 23 weeks after DM onset. Mice were fasted for 6 h, then they received an intraperitoneal injection of D-glucose solution (2 g/kg·body weight, Solarbio). Blood glucose levels were measured 0, 15, 30, 60, 90, and 120 min after the injection. Considering that the upper detection limit of the glucometer is 33.3 mmol/l, however, the blood glucose level of diabetic mice rises sharply after glucose injection. To be able to detect more accurate blood glucose levels in mice, we used the Blood Glucose Content Test Kit (Solarbio). We collected blood from the tail vein of mice using an anticoagulation tube and immediately centrifuged it to obtain serum. According to the instructions provided by the reagent vendor, we detected the blood glucose value by microplate reader.

### Insulin tolerance test (ITT)

ITT was performed 24 weeks post DM onset. After a 4-h fast, each mouse received an intraperitoneal injection of insulin (1 IU/kg·body weight). Blood glucose levels were monitored using a glucose meter (ACCU-CHEK Aviva, Roche, UK) 0, 15, 30, 60, 90, and 120 min after the injection.

### Echocardiography analysis

Twelve and twenty-four weeks after the establishment of DM, echocardiography was performed in M-mode and pulsed wave (PW) Doppler mode to assess the cardiac structure and function in mice, using a Vevo 3100LT echocardiography system (Vevo 3100LT, VisualSonics, SONICS, Newtown, Connecticut, USA). The mice were anesthetized using inhaled isoflurane (RWD). The concentration for induction of anesthesia was 2.0–2.5% and the concentration for maintenance of anesthesia was 1.5–1.8%, and the concentration was promptly adjusted according to the responsiveness of the mice. The mice were placed on the operating platform to monitor body temperature and heart rate. The chest hairs were removed using depilatory cream. LVEF, LVFS, and E/A ratio were obtained. All the measurements were carried out by two experienced technicians who were blind to the identities of the groups, and were averaged from three consecutive cardiac cycles.

### Assessment of cardiac pathology

After harvesting, the heart tissues were fixed into 10% neutral buffered formalin. After a 24-h fixation, the tissues were embedded in paraffin, followed by sectioning into 5-μm-thick sections. To assess cardiac histology and inflammatory cell infiltration, H&E staining was performed. For the evaluation of cardiac hypertrophy, heart sections were stained with WGA (GeneTex, Irvine, California, USA). The cross-sectional areas of the cardiomyocytes were observed by WGA staining of the hearts from mice per group, and evaluated by calculating the single myocyte cross-sectional areas measured by Image-Pro Plus software (Media Cybernetics, Silver Spring, USA). Cardiac fibrosis and apoptotic cell death were assessed by Masson's trichrome staining (Solarbio) and TUNEL staining (Beyotime, Shanghai, CHN). Image-Pro Plus software (Media Cybernetics, Silver Spring, USA) was used for quantification.

### Immunohistochemical staining

The heart tissues were fixed into 10% neutral buffered formalin. After a 24 h fixation, the tissues were embedded in paraffin and then sectioned into 5-μm-thick sections onto glass slides. Then the sections were incubated with primary antibodies at 4 °C overnight. After washing with PBS (5 min, 3 times, Servicebio, Hubei, Wuhan, CHN), the sections were incubated with a secondary antibody (BOSTER Biological Technology, Wuhan, Hubei, CHN) for 1 h at 37 °C. Then, a peroxidase substrate 3,3'-diaminobenzidine was used for color development. After counterstaining with hematoxylin, the sections were dehydrated and mounted in neutral balsam (Servicebio). Images were captured under OLYMPUS-BX53 upright fluorescence microscope (OLYMPUS, Beijing, CHN) or Pannoramic SCAN (3DHISTECH, Budapest, Hungary). And its positive area was quantified by Image-Pro Plus software (Media Cybernetics, Silver Spring, USA). The detailed information of the antibodies was provided in Reagents and Tools Table.

### Immunofluorescence staining

The heart tissues were placed in optimal cutting temperature (OCT, Servicebio) matrix compound at a temperature of −20 °C. Allow the heart to freeze for a minimum of 10 min, and then store at −80 °C. And then we sectioned them into 4-μm-thick sections onto glass slides. When we performed immunofluorescence double staining, on the first day, the sections were incubated with rabbit

polyclonal anti-F4/80 (1:100, proteintech, Cat# 29414-1-AP, RRID: AB_2918300, Wuhan, Hubei, CHN) at 4 °C overnight. On the second day, after washing with PBS (5 min, 3 times, Servicebio), the sections were incubated with a secondary antibody (Alexa Fluor® 488-conjugated Affinipure Goat Anti-Rabbit IgG; 1:500, Jackson ImmunoResearch Labs Cat# 111-545-003, RRID: AB_2338046, Philadelphia, Pennsylvania, USA) for 1 h at room temperature. Then we washed the sections with PBS three times every 5 min, the sections were incubated overnight at 4 °C with the following antibodies: mouse monoclonal anti-iNOS (1:100, Santa Cruz Biotechnology Cat# sc-7271, RRID: AB_627810), mouse monoclonal anti-CD86 (1:100, Santa Cruz Biotechnology Cat# sc-28347, RRID: AB_627200), mouse monoclonal anti-CD206 (1:100, Santa Cruz Biotechnology Cat# sc-58986, RRID: AB_2144945), mouse monoclonal anti-HO-1 (1:100, Proteintech Cat# 66743-1-Ig, RRID: AB_2882091), mouse monoclonal anti-TrxR1 (1:100, Santa Cruz Biotechnology Cat# sc-28321, RRID: AB_628405). On the third day, consistent with the above steps, after the secondary antibody (Alexa Fluor® 594-conjugated Affinipure Goat Anti-Mouse IgG; 1:500, Jackson ImmunoResearch Labs Cat# 115-585-003, RRID: AB_2338871) was incubated, we performed DAPI (Beyotime) staining of the sections. Finally, we sealed the sections with an anti-fluorescence quenching sealer. Images were captured under Leica TCS SP8 confocal fluorescence microscope (Leica Microsystems, Shanghai, CHN).

### Detection of reactive oxygen species (ROS)

For the detection of cardiac and cellular ROS, the sections, HL-1 cells and RAW264.7 cells were incubated with dihydroethidium (DHE, 1 μmol/l, Beyotime) solution at 37 °C for 30 min.

### RNA extraction and quantitative real-time PCR (qRT-PCR) analysis

Total RNA was extracted with E.Z.N.A.®Total RNA Kit I (Omega BIO-TEK, Norcross, Georgia, USA) and reversely transcribed to cDNA using a PrimeScript RT reagent Kit (Takara Biomedical Technology, Beijing, CHN). The qRT-PCR was performed using LightCycler 480 SYBR Green I Master (Roche, Shanghai, CHN) on a QuantStudio 5 Real-Time PCR System (Thermo Fisher Scientific). Relative quantification for gene expression was calculated using the $2^{-\Delta\Delta Ct}$ method. *Gapdh* was used as an internal control. The primer sequences were summarized in Appendix Table S3. And all primers were provided by Sangon Biotech (Shanghai, CHN).

### Western blot

Cardiac tissues and cultured HL-1 cardiomyocytes and RAW264.7 cells were collected and homogenized with RIPA buffer, which contained protease inhibitor and phosphatase inhibitor (all from Beyotime). The denatured proteins were separated by gel electrophoresis, transferred to a poly-vinylidene fluoride (PVDF) membrane, and blocked with 5% (wt/vol) skim milk (Solarbio). The membrane was incubated overnight with primary antibodies at 4 °C and were incubated with secondary antibody for 1 h at room temperature. All the primary antibodies were diluted with primary antibody diluent (Beyotime), and secondary antibodies were diluted with TBST (Servicebio). A chemiluminescence system (Tanon, Shanghai, CHN) was used for analyzing immunoreactive bands. Western blot images were quantified utilizing Image Studio™ Lite software (LI-COR, Lincoln, Nebraska, USA). The detailed information of the antibodies was provided in Reagents and Tools Table.

### Statistical analysis

At least 3 mice per group were studied. Cell experiments were performed in triplicate. Human macrophages were obtained from six T2DM patients per group. The Data are presented as the means ± SEM. Two-group comparisons were analyzed by a two-tailed unpaired Student's t-test and more than two-group comparisons were analyzed by one-way ANOVA. The Mann–Whitney U test was used to predict whether there was a significant difference in clinical information between groups of human subjects. $P < 0.05$ was considered statistically significant. Analyses were performed using the SPSS 26.0 or GraphPad Prism 9.0 software.

### Graphics

BioRender was used to prepare the synopsis of this study.

## Data availability

The data supporting the findings of this study are available in the manuscript and its supplementary information. The RNA-seq data for Fig. 5A and EV4 are available via NCBI SRA Data with identifier PRJNA1007889. It is accessible via this weblink: https://www.ncbi.nlm.nih.gov/bioproject/1007889. The RNA-seq data for Appendix Figs. S1 and S2 are available via NCBI SRA Data with identifier PRJNA1007927. It is accessible via this weblink: https://www.ncbi.nlm.nih.gov/bioproject/1007927.

The source data of this paper are collected in the following database record: biostudies:S-SCDT-10_1038-S44319-024-00271-x.

## Peer review information

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

## Acknowledgements

We thank all for the individuals participated in this work. This work was supported in part by the National Natural Science Foundation of China [81973031] and Cheeloo Young Scholar Program of Shandong University [21320089963054] to Hao Wu, and Natural Science Foundation of Shandong Province [ZR2021MH204] to Xiao Yin.

## Author contributions

**Jie Li**: Data curation; Formal analysis; Validation; Investigation; Visualization; Methodology; Writing—original draft. **Wenshan Nan**: Data curation; Formal analysis; Validation; Investigation; Visualization; Methodology; Writing—original draft. **Xiaoli Huang**: Data curation; Writing—review and editing. **Huali Meng**: Data curation; Writing—review and editing. **Shue Wang**: Data curation; Writing—review and editing. **Yan Zheng**: Data curation; Writing—review and editing. **Ying Li**: Data curation; Writing—review and editing. **Hui Li**: Supervision; Writing—review and editing. **Zhiyue Zhang**: Supervision; Writing—review and editing. **Lei Du**: Supervision; Writing—review and editing. **Xiao Yin**: Resources; Supervision; Funding acquisition; Project administration; Writing—review and editing. **Hao Wu**: Conceptualization; Resources; Supervision; Funding acquisition; Writing—original draft; Project administration; Writing—review and editing.

Source data underlying figure panels in this paper may have individual authorship assigned. Where available, figure panel/source data authorship is listed in the following database record: biostudies:S-SCDT-10_1038-S44319-024-00271-x.

## Disclosure and competing interests statement

The authors declare no competing interests.

# Expanded View Figures

**Figure EV1.  EPA lowered fasting blood glucose level, and improved glucose tolerance and insulin sensitivity in the diabetic mice.**

(**A**) Schematic diagram for the induction of DM and EPA treatment. (**B**) Fasting blood glucose levels. At the 20$^{th}$ week, \*\*\*$P < 0.001$, DM vs. Ctrl; $^†P = 0.015$, DM/EPA vs. DM. At the 24th week, \*\*\*$P < 0.001$, DM vs. Ctrl; $^{†††}P < 0.001$, DM/EPA vs. DM. (**C**) Body weight of the mice. Body weight was recorded every week, and presented every 2 weeks. At the 16th week, \*$P = 0.044$, DM vs. Ctrl. At the 18th week, \*$P = 0.027$, DM vs. Ctrl; $^†P = 0.011$, DM/EPA vs. DM. At the 20th week, \*\*$P = 0.003$, DM vs. Ctrl. At the 22th week, \*$P = 0.041$, DM vs. Ctrl; $^†P = 0.032$, DM/EPA vs. DM. At the 24th week, \*$P = 0.026$, DM vs. Ctrl. (**D, E**) Curve of blood glucose levels during GTT, and area under the curve. For (**D**), at the start, \*\*\*$P < 0.001$, DM vs. Ctrl; $^{††}P = 0.002$, DM/EPA vs. DM. At the 15th minute, \*\*\*$P < 0.001$, DM vs. Ctrl. At the 30th minute, \*\*\*$P < 0.001$, DM vs. Ctrl; $^{††}P = 0.009$, DM/EPA vs. DM. At the 60th minute, \*\*\*$P < 0.001$, DM vs. Ctrl; $^{††}P = 0.001$, DM/EPA vs. DM. At the 90th minute, \*\*\*$P < 0.001$, DM vs. Ctrl; $^{††}P = 0.001$, DM/EPA vs. DM. At the 120th minute, \*\*\*$P < 0.001$, DM vs. Ctrl; $^{†††}P < 0.001$, DM/EPA vs. DM. For (**E**), \*\*\*$P < 0.001$, DM vs. Ctrl; $^{†††}P < 0.001$, DM/EPA vs. DM. (**F, G**) Curve of blood glucose levels during ITT, and area under the curve. For (**F**), at the start, \*\*\*$P < 0.001$, DM vs. Ctrl; $^{†††}P < 0.001$, DM/EPA vs. DM. At the 15th minute, \*\*\*$P < 0.001$, DM vs. Ctrl. $^†P = 0.01$, DM/EPA vs. DM. At the 30th, 60th, 90th, and 120th minute, \*\*\*$P < 0.001$, DM vs. Ctrl; $^{†††}P < 0.001$, DM/EPA vs. DM. For (**G**), \*\*\*$P < 0.001$, DM vs. Ctrl; $^{†††}P < 0.001$, DM/EPA vs. DM. Data information: Data are represented as individual data points of $n = 10, 14, 14$ (**B**), $n = 10, 14, 14$ (**C**), $n = 10, 10, 10$ (**D, E**), $n = 7, 7, 7$ (**F, G**) biological replicates and means ± SEM. Analysis by one-way ANOVA. AUC area under curve, EPA eicosapentaenoic acid, FBG fasting blood glucose, GTT glucose tolerance test, HFD high-fat diet, ITT insulin tolerance test, STZ streptozotocin. Groups: Ctrl control, DM diabetes mellitus, DM/EPA diabetic mice supplemented with EPA.

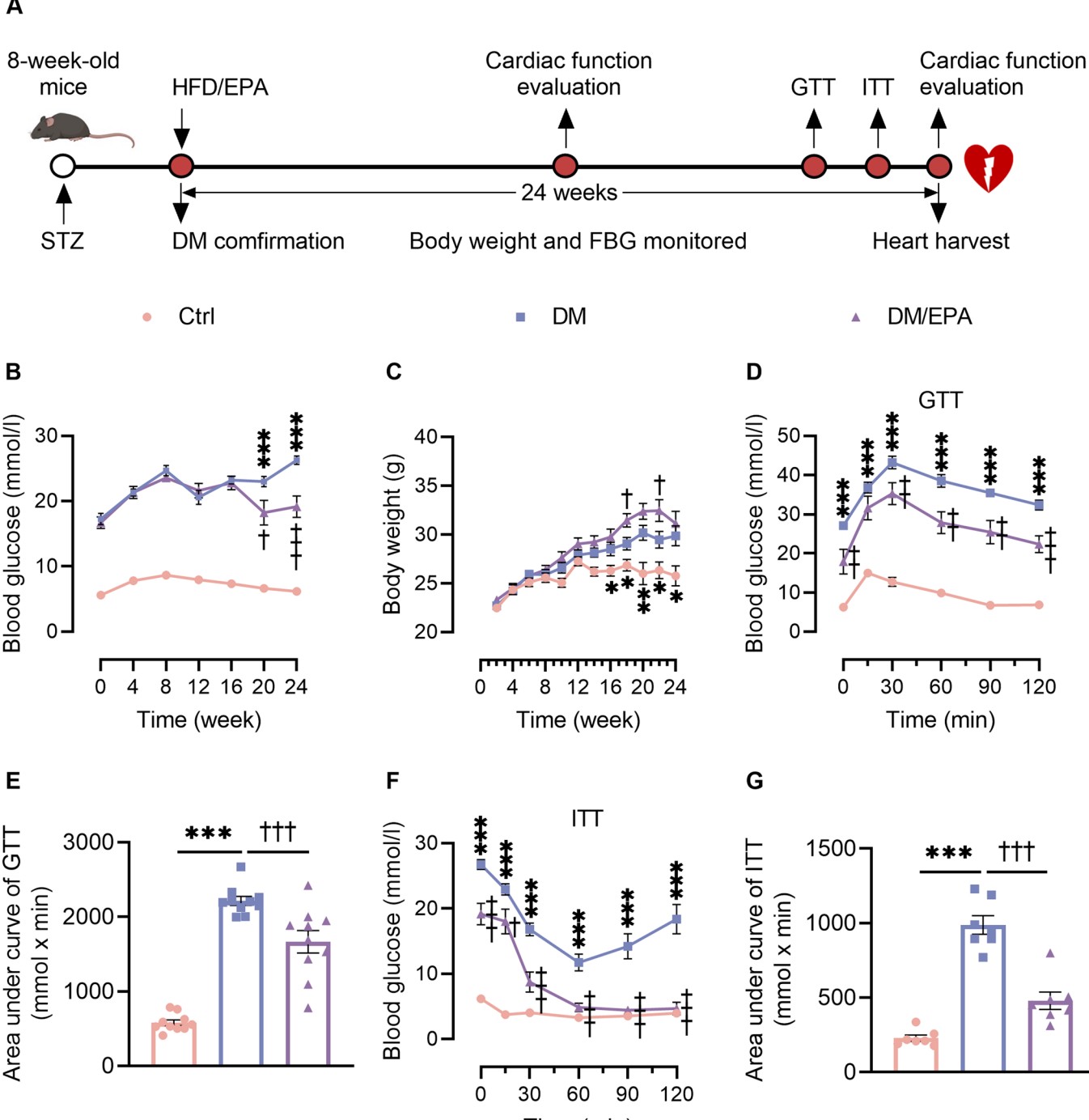

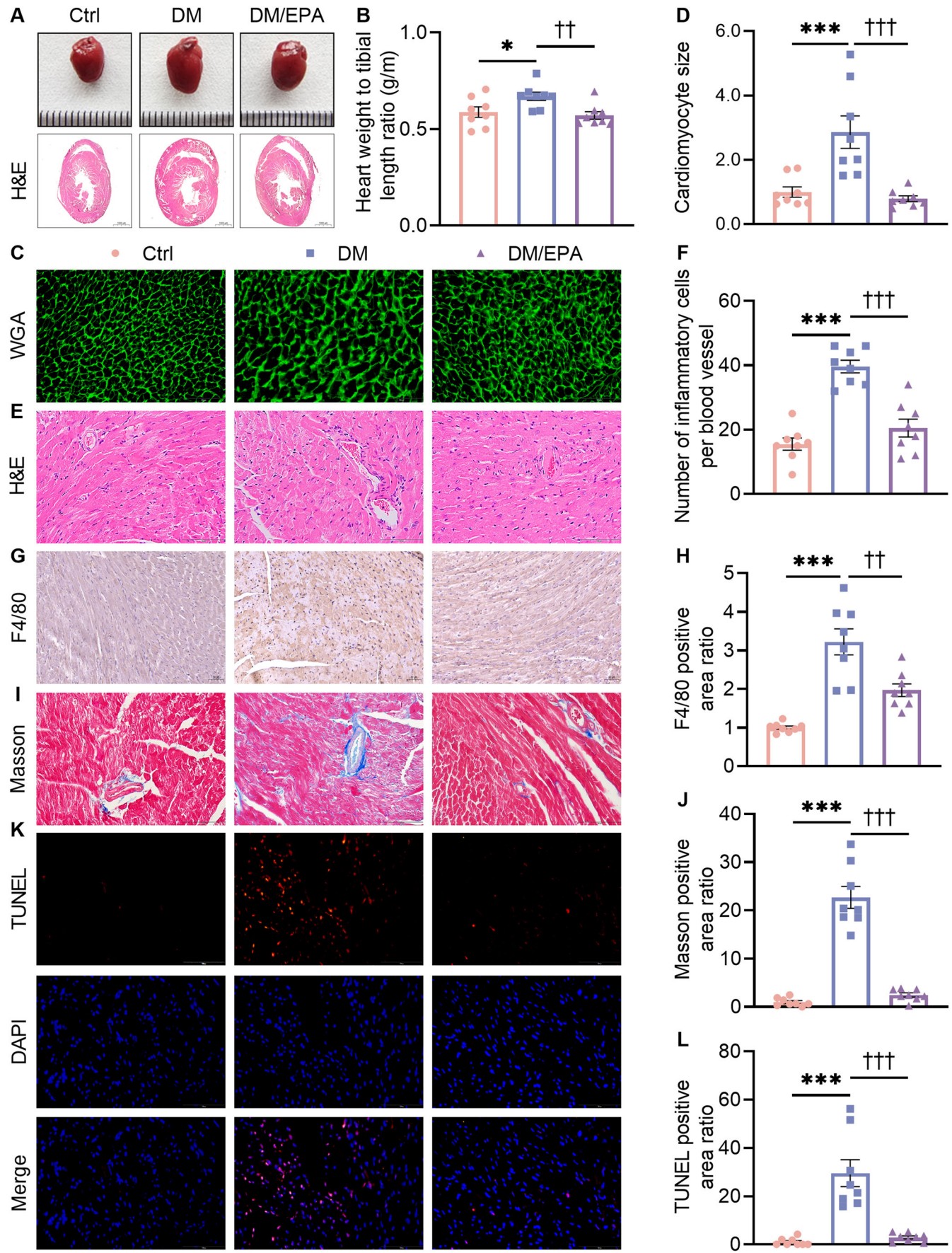

◀ **Figure EV2. EPA prevented DM-induced cardiac pathological injuries.**

(A) Representative gross images of the entire hearts (upper panel, bar = 1 mm), and cross-section images of H&E staining of the hearts (lower panel, bar = 1000 μm). (B) The ratio of heart-weight-to-tibia length. *$P$ = 0.021, DM vs. Ctrl; ††$P$ = 0.006, DM/EPA vs. DM. (C, D) WGA staining (bar = 100 μm) and quantification of cardiac hypertrophy. For (D), ***$P$ < 0.001, DM vs. Ctrl; †††$P$ < 0.001, DM/EPA vs. DM. (E, F) H&E staining (bar = 100 μm) and the number of inflammatory cells infiltrated per vessel. For (F), ***$P$ < 0.001, DM vs. Ctrl; †††$P$ < 0.001, DM/EPA vs. DM. (G, H) Immunohistochemical staining of F4/80 and quantification of positive area (bar = 50 μm). For (H), ***$P$ < 0.001, DM vs. Ctrl; ††$P$ = 0.001, DM/EPA vs. DM. (I, J) Masson's trichrome staining (bar = 100 μm) with the positive area quantified. For (J), ***$P$ < 0.001, DM vs. Ctrl; †††$P$ < 0.001, DM/EPA vs. DM. (K, L) TUNEL assay (bar = 100 μm) by double staining with DAPI (blue) and TUNEL (red), and ratio of the TUNEL positive nuclei. For (L), ***$P$ < 0.001, DM vs. Ctrl; †††$P$ < 0.001, DM/EPA vs. DM. Data information: For (D, H, J, L), the data are normalized to Ctrl. Data are represented as individual data points of $n$ = 8 (B, D, F, H, J, L) biological replicates and means ± SEM. Analysis by one-way ANOVA. DAPI 4′,6-diamidino-2-phenylindole, TUNEL terminal deoxynucleotidyl transferase (TdT)-mediated dUTP nick-end labeling, WGA wheat germ agglutinin. Other abbreviations are the same as in Fig. EV1. Groups: Ctrl control, DM diabetes mellitus, DM/EPA diabetic mice supplemented with EPA.

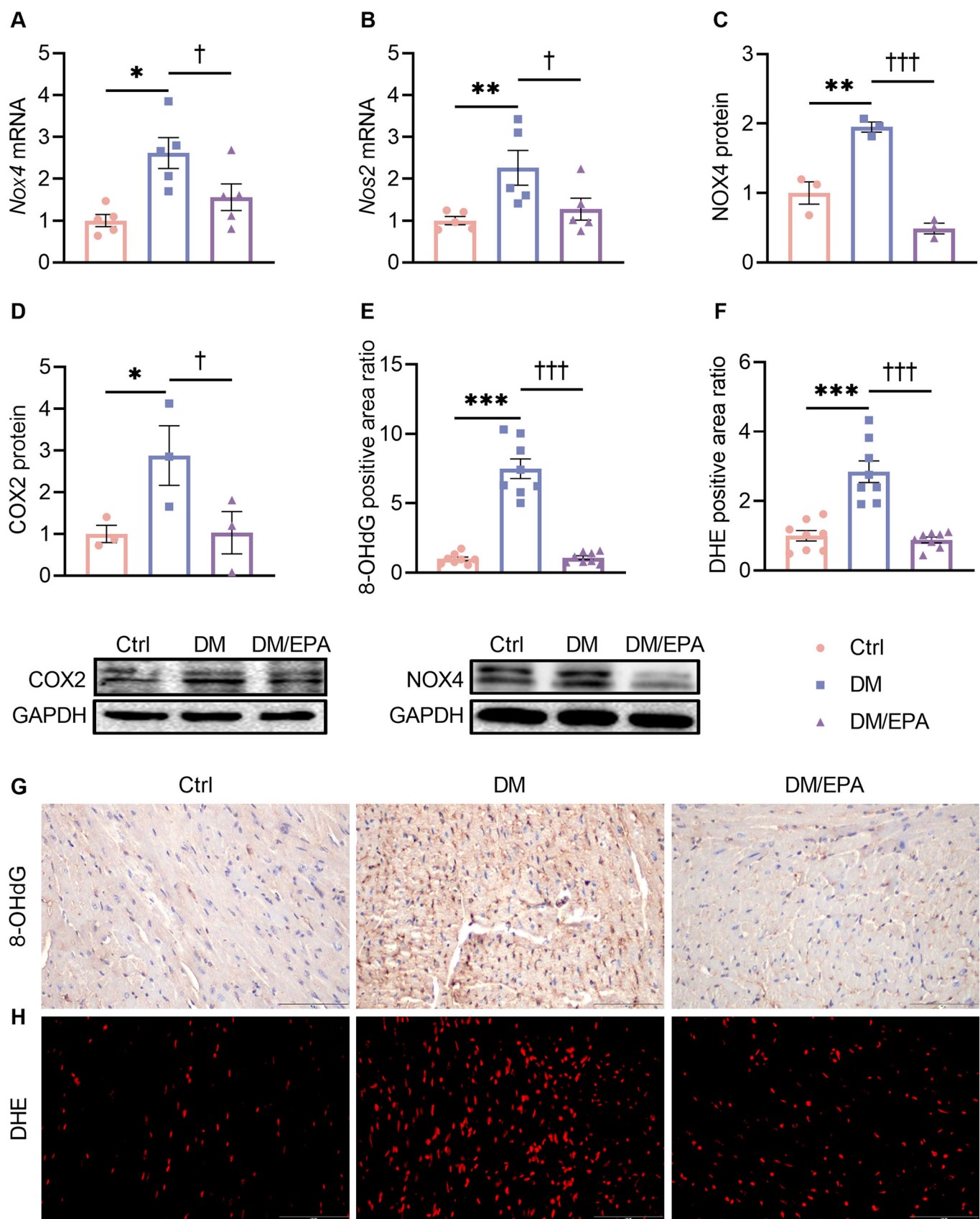

◀  **Figure EV3.  EPA attenuated DM-induced cardiac oxidative stress.**

(**A**, **B**) mRNA expression of *Nox4* and *Nos2* were determined by qRT-PCR. For (**A**), *$P$ = 0.02, DM vs. Ctrl; †$P$ = 0.025, DM/EPA vs. DM. For (**B**), **$P$ = 0.009, DM vs. Ctrl; †$P$ = 0.032, DM/EPA vs. DM. (**C**, **D**) Protein levels of NOX4 and COX2 were determined by Western blot. For (**C**), **$P$ = 0.001, DM vs. Ctrl; †††$P$ < 0.001, DM/EPA vs. DM. For (**D**), *$P$ = 0.043, DM vs. Ctrl; †$P$ = 0.045, DM/EPA vs. DM. (**E**) Quantification of 8-OHdG positive area of the hearts. ***$P$ < 0.001, DM vs. Ctrl; †††$P$ < 0.001, DM/EPA vs. DM. (**F**) Quantification of DHE positive area of the hearts. ***$P$ < 0.001, DM vs. Ctrl; †††$P$ < 0.001, DM/EPA vs. DM. (**G**) Immunohistochemical staining of 8-OHdG (bar = 100 μm). (**H**) DHE staining of heart sections (bar = 100 μm). Data information: For (**A–F**), the data are normalized to Ctrl. Data are represented as individual data points of $n$ = 5, 5, 5 (**A**, **B**), $n$ = 3, 3, 3 (**C**, **D**), $n$ = 8, 8, 8 (**E**), $n$ = 8, 8, 8 (**F**) biological replicates and means ± SEM. Analysis by one-way ANOVA. 8-OHdG 8-hydroxy-2′-deoxyguanosine, COX2 cyclooxygenase-2, DHE dihydroethidium, *Nos2* nitric oxide synthase 2, *Nox4*/NOX4 NADPH oxidase 4. Other abbreviations are the same as in Fig. EV1. Groups: Ctrl control, DM diabetes mellitus, DM/EPA diabetic mice supplemented with EPA.

**A**

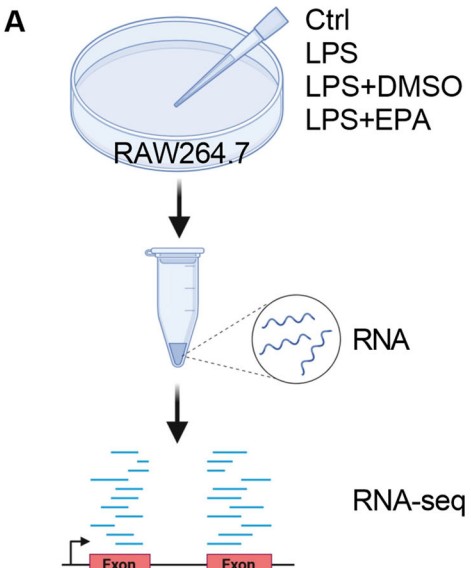

**B**

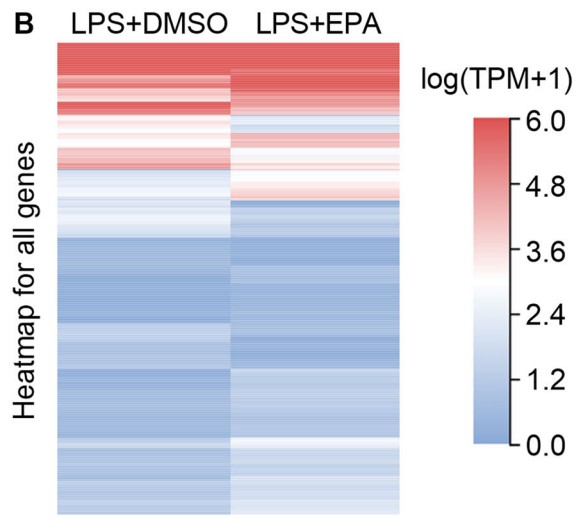

**C**

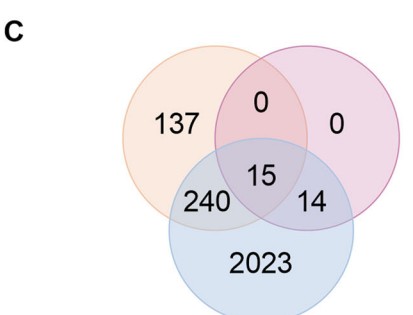

LPS+DMSO vs. LPS+EPA
LPS vs. LPS+DMSO
Ctrl vs. LPS

**D**

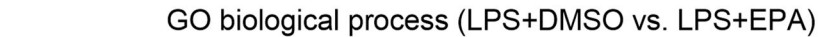
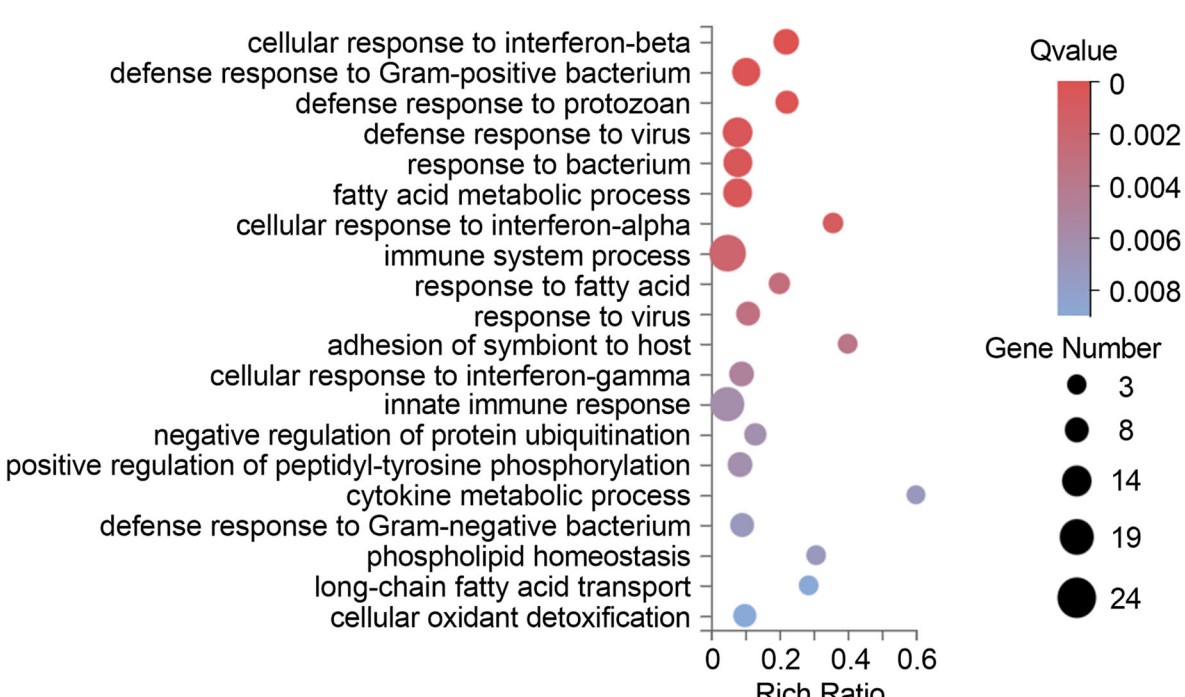

GO biological process (LPS+DMSO vs. LPS+EPA)

◀ **Figure EV4.** **EPA regulated immune-associated signal pathways in M1-polarized macrophages.**

(A) Schematic representation of the RNA-seq protocol. (B) Heat map comparison of the global mRNA expression profiles in LPS + DMSO and LPS + EPA groups. (C) Venn diagram showing comparisons of altered gene numbers among the groups. (D) GO analysis for top biological processes altered by EPA compared with DMSO in LPS-challenged RAW264.7 cells. GO Gene Ontology, LPS lipopolysaccharides, RNA-seq RNA-sequencing. Other abbreviations are the same as in Fig. EV1. Groups: Ctrl control, LPS LPS-stimulated RAW264.7 cells, LPS + DMSO LPS-stimulated RAW264.7 cells treated with DMSO, LPS + EPA LPS-stimulated RAW264.7 cells treated with EPA.

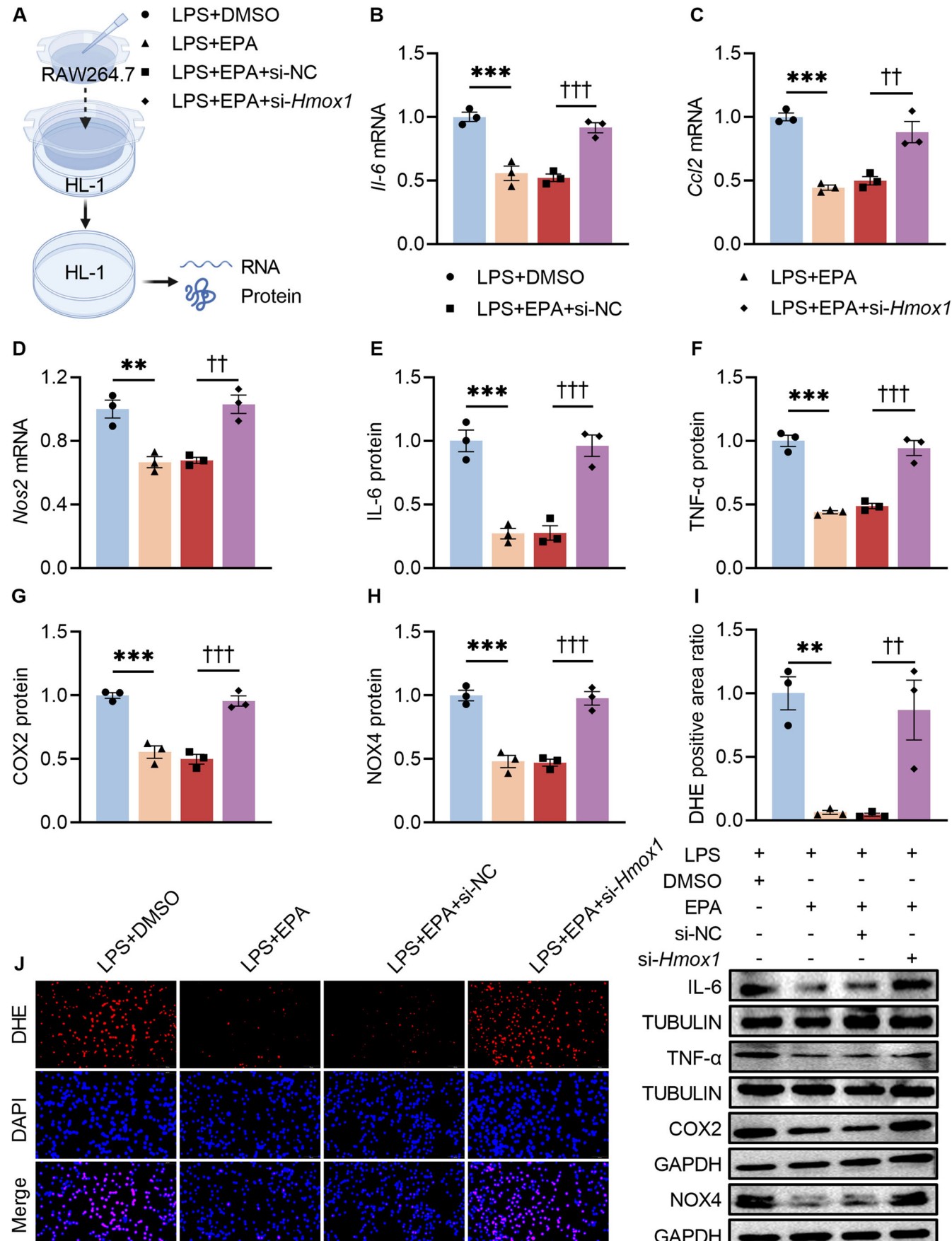

◄ **Figure EV5. HO-1 was required for EPA repression of cardiomyocyte injury induced by M1 polarized macrophages.**

(A) Schematic representation of the experimental protocol. (B–D) mRNA levels of *Il-6*, *Ccl2*, and *Nos2* in HL-1 cells. For (B), \*\*\*P < 0.001, LPS + EPA vs. LPS + DMSO; ^†††^P < 0.001, LPS + EPA+si-*Hmox1* vs. LPS + EPA+si-NC. For (C), \*\*\*P < 0.001, LPS + EPA vs. LPS + DMSO; ^††^P = 0.001, LPS + EPA+si-*Hmox1* vs. LPS + EPA+si-NC. For (D), \*\*P = 0.001, LPS + EPA vs. LPS + DMSO; ^††^P = 0.001, LPS + EPA+si-*Hmox1* vs. LPS + EPA+si-NC. (E–H) Protein levels of IL-6, TNF-α, COX2, and NOX4. \*\*\*P < 0.001, LPS + EPA vs. LPS + DMSO; ^†††^P < 0.001, LPS + EPA+si-*Hmox1* vs. LPS + EPA+si-NC. (I, J) DHE assay (bar = 100 μm) by double staining with DAPI (blue) and DHE (red), and ratio of DHE positive cells. For (I), \*\*P = 0.001, LPS + EPA vs. LPS + DMSO; ^††^P = 0.003, LPS + EPA+si-*Hmox1* vs. LPS + EPA+si-NC. Data information: For (B–I), the data are normalized to LPS + DMSO. Data are represented as individual data points of n = 3 (B–I) biological replicates and means ± SEM. Analysis by one-way ANOVA. *Ccl2* C-C motif chemokine ligand 2, *Il-6*/IL-6 interleukin-6, si-*Hmox1* *Hmox1* siRNA, si-NC negative control siRNA, TNF-α tumor necrosis factor-α. Other abbreviations are the same as in Figs. EV1–EV4. Groups: LPS + DMSO HL-1 cells co-cultured with LPS-stimulated RAW264.7 cells and treated with DMSO, LPS + EPA HL-1 cells co-cultured with LPS-stimulated RAW264.7 cells and treated with EPA, LPS + EPA+si-NC LPS + EPA treated with si-NC, LPS + EPA+si-*Hmox1* LPS + EPA treated with si-*Hmox1*.

