## [Peer Review File · EMBO Reports]

Eicosapentaenoic acid induces macrophage Mox polarization to prevent diabetic cardiomyopathy

Jie Li, Wenshan Nan, Xiaoli Huang, Huali Meng, Shue Wang, Yan Zheng, Ying Li, Hui Li, Zhiyue Zhang, Lei Du, Xiao Yin, and Hao Wu

Corresponding author(s): Hao Wu (hwu@sdu.edu.cn) , Xiao Yin (yinxiao@sdu.edu.cn)

Review Timeline:

Submission Date:	7th Mar 24
Editorial Decision:	19th Apr 24
Revision Received:	9th Jun 24
Editorial Decision:	13th Aug 24
Revision Received:	21st Aug 24
Accepted:	15th Sep 24

Editor: Deniz Senyilmaz Tiebe

Transaction Report:

Dear Prof. Wu,

Thank you for submitting your preliminary point-by-point response. I have now looked at your points carefully. I appreciate that you can address many of the concerns raised and see that the proposed experiments will strengthen the manuscript.

Having looked at everything, I would like to invite you to submit a revised manuscript. However, I would like to point out that we need strong support from the referees to consider publication here. Please revise your manuscript with the understanding that the referee concerns (as in their reports) must be fully addressed and their suggestions taken on board. Please address all referee concerns in a complete point-by-point response. Acceptance of the manuscript will depend on a positive outcome of a second round of review. It is EMBO reports policy to allow a single round of major experimental revision only and acceptance or rejection of the manuscript will therefore depend on the completeness of your responses included in the next, final version of the manuscript.

We realize that it is difficult to revise to a specific deadline. In the interest of protecting the conceptual advance provided by the work, we recommend a revision within 3 months. Please discuss the revision progress ahead of this time with me if you require more time to complete the revisions, or if you have questions or comments regarding the revision (also by video chat).

1. A data availability section providing access to data deposited in public databases is missing (where applicable).
2. Your manuscript contains statistics and error bars based on $n=2$. Please use scatter plots in these cases.

You can submit the revision either as a Scientific Report or as a Research Article. For Scientific Reports, the revised manuscript can contain up to 5 main figures and 5 Expanded View figures, and it should not exceed 27000 characters. If the revision leads to a manuscript with more than 5 main figures it will be published as a Research Article. In this case the Results and Discussion section should be separate. If a Scientific Report is submitted, these sections have to be combined. This will help to shorten the manuscript text by eliminating some redundancy that is inevitable when discussing the same experiments twice. In either case, all materials and methods should be included in the main manuscript file.

4) a .docx formatted letter INCLUDING the reviewers' reports and your detailed point-by-point responses to their comments. As part of the EMBO publication's Transparent Editorial Process, EMBO reports publishes online a Review Process File (RPF) to accompany accepted manuscripts. This File will be published in conjunction with your paper and will include the referee reports, your point-by-point response and all pertinent correspondence relating to the manuscript.

<https://www.embopress.org/page/journal/14693178/authorguide#transparentprocess>

5) a complete author checklist, which you can download from our author guidelines <https://www.embopress.org/page/journal/14693178/authorguide>. Please insert information in the checklist that is also reflected in the manuscript. The completed author checklist will also be part of the RPF.

6) Please note that all corresponding authors are required to supply an ORCID ID for their name upon submission of a revised manuscript (<<https://orcid.org/>>). Please find instructions on how to link your ORCID ID to your account in our manuscript tracking system in our Author guidelines <<https://www.embopress.org/page/journal/14693178/authorguide#authorshipguidelines>>

Additional information on source data and instruction on how to label the files are available: <https://www.embopress.org/page/journal/14693178/authorguide#sourcedata>

9) Our journal encourages inclusion of *data citations in the reference list* to directly cite datasets that were re-used and obtained from public databases. Data citations in the article text are distinct from normal bibliographical citations and should directly link to the database records from which the data can be accessed. In the main text, data citations are formatted as follows: "Data ref: Smith et al, 2001" or "Data ref: NCBI Sequence Read Archive PRJNA342805, 2017". In the Reference list, data citations must be labeled with "[DATASET]". A data reference must provide the database name, accession number/identifiers and a resolvable link to the landing page from which the data can be accessed at the end of the reference. Further instructions are available at <http://www.embopress.org/page/journal/14693178/authorguide#referencesformat>

- the name of the statistical test used to generate error bars and P values,
- the number (n) of independent experiments (please specify technical or biological replicates) underlying each data point,
- the nature of the bars and error bars (s.d., s.e.m.),
- If the data are obtained from n Program fragment delivered error ``Can't locate object method "less" via package "than" (perhaps you forgot to load "than"?) at //ejpvfs23/sites23b/embo/www/letters/embo_decision_revise_and_review.txt line 56.' 2, use scatter blots showing the individual data points.

12) Please also note our reference format: <http://www.embopress.org/page/journal/14693178/authorguide#referencesformat>

We would also welcome the submission of cover suggestions, or motifs to be used by our Graphics Illustrator in designing a

cover.

I look forward to seeing a revised version of your manuscript when it is ready. Please let me know if you have questions or comments regarding the revision.

Kind regards,

Deniz Senyilmaz Tiebe

Deniz Senyilmaz Tiebe, PhD
Scientific Editor
EMBO Reports

Referee #1:

In this manuscript by Li et al. the authors investigate the impact of eicosapentaenoic acid on macrophage Mox polarization in diabetic cardiomyopathy.

I have the following some comments.

- 1) DCM is the abbreviation of dilated cardiomyopathy, not of diabetic cardiomyopathy.
- 2) M1 polarization was induced in RAW264.7 cells via LPS and the impact of EPA on M1 polarization was evaluated. What is known about the impact of hyperglycemia with/out FFA on macrophage polarization?
- 3) How do the authors explain the different n number depending of the analysis? How was the group size of the animals determined?
- 4) In vitro data show data of 3 independent experiments, resulting in 3 data points. Though, how many technical replicates were used per independent experiment? This information is necessary.
- 5) Indirect information is provided about macrophage polarization in DM mice treated with EPA. Flow cytometry and/or immunofluorescence is recommended to evaluate macrophage subsets in the heart.
- 6) It is strongly recommended to have the manuscript proofread by a native English speaker.

Referee #2:

In this study, the authors provide data suggesting that omega-3 fatty acid, EPA prevents diabetic cardiomyopathy through inducing Mox macrophage polarization. Given that many other evidence of EPA prevent metabolic syndromes including type 2 diabetes, obesity-induced insulin resistance, one more beneficial role of EPA in diabetic cardiomyopathy is quite expectable. However, there is not much of studies how EPA prevents various adverse outcomes from metabolic syndromes including diabetic cardiomyopathy as authors stated. This is an interesting study, revealing EPA modulates macrophage polarization into less inflamed states in diabetic cardiomyopathy, but not conceptually novel or thought-provoking. Authors also claim that EPA-induced heme oxygenase-1 is a keystone of Mox macrophage-mediated anti-DCM phenotype, but HO-1 induction is triggered by a variety of stimuli, thus the direct link between EPA and HO-1 in anti-DCM effects need to be validated in depth mechanistic studies.

1. Based on authors data, EPA does prevent diabetic cardiomyopathy in vivo and in vitro experiments and HO-1 is involved in these phenotypes. However, HO-1 induction by other stimuli may lead to similar phenotype without EPA treatment. EPA may need HO-1 to show these beneficial phenotypes, but HO-1 induction (stronger) could be achieved by other stimuli and lead to better anti-DCM effects.
2. How does EPA modulate Mox polarization? What can be the possible molecular mechanism? Is there a particular membrane sensor for EPA and then transduce signaling to activate HO-1 in macrophages to polarize into Mox?

Referee #3:

Major Comments:

1. The authors only use H&E staining to make general claims about immune cell infiltration; yet stain for F4/80 by confocal microscopy is used in other panels of the manuscript. Quantification of F4/80+ cells per area should be added to solidify the claims.
2. iNOS staining as an M1 marker in Figure 2 I/J appears diffuse. Maybe use better representative images or more convincingly include an additional M1 surface marker to co-localize with F4/80?

3. The authors make use of the HL-1 cell line throughout the manuscript, yet previous work has demonstrated that HL-1 cells have impaired ability to shift metabolism in response to injurious stimuli (Kuznetsov et al. 2015 BBA). Please explain the rationale for using this cell type and/or include a statement in a limitations section?
4. The authors make claims about the effect of EPA not impacting M2 polarization. Please include a description of the method how RAW cells are polarized to an M2 phenotype. Also please clarify the experimental set-up for this experiment. Furthermore, this data seems to be missing from the manuscript (i.e. figure S3 is missing).
5. The authors need to confirm by RT-qPCR and Western blotting that Hmox1 siRNA effectively knocks down HMOX1 expression.
6. Please clarify how cardiomyocyte size was measured as this is not clear from the methods or the figure.
7. The authors report decreased heart size, but also concurrent increases in body weight. In this case it would be appropriate to normalize heart mass to tibial length or some other standard which is unchanged in the DM mice.
8. Please clarify whether perivascular Masson's trichrome staining was included or excluded for quantification.

Minor Comments:

1. Be consistent with references to supplemental figures throughout the text.
2. Labels should be included on the GTT and ITT graphs in supplementary figure 1.
3. Are other NOX isoforms (in addition to NOX4) changed in the HL-1 co-culture experiments?

June 9, 2024

Dear Prof. Tiebe,

Thank you very much for your consideration of our manuscript “**Eicosapentaenoic acid induces macrophage Mox polarization in prevention of diabetic cardiomyopathy**” (EMBOR-2024-59157)! We appreciate the comments from the reviewers. In particular, we agree with you and the referee #2 for your suggestion on conducting an investigation of the mechanism by which EPA induced HO-1 expression. We would like to perform further experiments on this point. In addition, we are also grateful for other comments by the referees, and have made point-to-point response to these comments (Please find them below):

Referee #1:

Comment 1: DCM is the abbreviation of dilated cardiomyopathy, not of diabetic cardiomyopathy.

Response: Thank you very much for your reminder! We have carefully considered this suggestion, and changed “DCM” to “DC” to ensure that the abbreviation is more accurate.

Comment 2: M1 polarization was induced in RAW264.7 cells via LPS and the impact of EPA on M1 polarization was evaluated. What is known about the impact of hyperglycemia with/out FFA on macrophage polarization?

Response:

Thank you very much for your question! Hyperglycemia (HG) with/out FFA has been shown to affect macrophage polarization *in vitro* and *in vivo* in previous studies.

The effects of HG on macrophage polarization

Generally, HG promoted the production of M1 polarized macrophages and associated inflammatory factors while inhibiting M2 polarized macrophages. The study showed that hyperglycemic environment can lead to increased pro-inflammatory cytokines (Morey *et al.*, 2019), which encouraged the vicious cycle of maintaining M1 macrophage polarization and chronic inflammation (Morey *et al.*, 2019; Pavlou *et al.*, 2018). For instance, HG was found to stimulate acute production of TNF- α and long-term production of IL-1 β during macrophage differentiation (Moganti *et al.*, 2017). Similarly, mouse peritoneal macrophages treated with hyperglycemic medium showed an increase in expression of pro-inflammatory cytokines including IL-1 β , IL-6 and TNF- α in a time and dose-dependent manner (Wen *et al.*, 2006). Furthermore, HG-induced reactive oxygen species may favor induction of M1 pro-inflammatory macrophages during diabetes onset and progression (Rendra *et al.*, 2019).

HG showed inhibitory effects on M2 polarization of macrophage. Studies revealed that both the serum concentration of IL-10 in prediabetic patients (fasting blood glucose levels about 6 mM) as well as the IL-10 production by HG treated human monocyte-derived macrophages was decreased (Torres-Castro *et al.*, 2016). In addition, RAW264.7 macrophages and mouse bone marrow-derived macrophages (BMDM) exposed to HG were resistant to anti-inflammatory action of IL-10 (Barry *et*

al, 2016) . Furthermore, HG aggravated hepatic ischemia and reperfusion injury by inhibiting liver-resident macrophage M2 polarization (Rao *et al*, 2017).

The effects of FFA on macrophage polarization

Similar to HG, FFA exhibited a dual effect of promoting M1 and inhibiting M2 polarization of macrophage. In obese adipose tissue local pro-inflammatory microenvironment, the presence of FFAs contributed to adipocytes activation and led to the differentiation of M1 macrophages (Feng *et al*, 2017; Weisberg *et al*, 2003). Treatment of BMDM from high fat diet-fed mice with palmitic acid (PA) significantly promoted M1 macrophages, as indicated by increased proinflammatory cytokines, along with decreased M2 macrophages markers (Zhou *et al*, 2023).

The effects of HG with FFA on macrophage polarization

It was found that the relative mRNA expression of CD86/CD206 were significantly increased in HG/PA (20 mM/0.3 mM)-stimulated RAW264.7 cells (Liu *et al*, 2023), which suggested that HG and FFA jointly promote macrophage M1 polarization. HG and elevated FFA levels often coexist in conditions like obesity and type 2 diabetes. The combination of HG and FFA can have synergistic effects on macrophage polarization, exacerbating inflammation and insulin resistance. The crosstalk between glucose and lipid metabolism pathways in macrophages contributed to their polarization towards the M1 phenotype under conditions of metabolic dysfunction (Chawla *et al*, 2011; Thapa & Lee, 2019).

Comment 3: How do the authors explain the different n number depending of the analysis? How was the group size of the animals determined?

Response: Thank you very much for this comment! At the onset of the animal experiments, the n number of animals in the Ctrl, DM and DM/EPA group were 10, 10, and 15 respectively. Due to varying experimental needs, the n number of animals or heart tissues allocated to each experiment differed. Additionally, some mice had poor baseline physiological indicators, were unsuitable for the experiment, or died during the process of the experiment, leading to variations in the final n numbers. Despite this, we ensured statistical significance by maintaining $n \geq 3$ in each group.

Comment 4: In vitro data show data of 3 independent experiments, resulting in 3 data points. Though, how many technical replicates were used per independent experiment? This information is necessary.

Response: Thank you very much for your reminder! All data are represented as individual data points biological replicates and mean \pm SEM, and three technical replicates were used per independent experiment.

Comment 5: Indirect information is provided about macrophage polarization in DM mice treated with EPA. Flow cytometry and/or immunofluorescence is recommended to evaluate macrophage subsets in the heart.

Response: Thank you very much for your reminder! We evaluated M1 and Mox polarization of macrophages in heart tissues of diabetic mice. According to the recommendation of referee #1, we evaluated the M2 polarization of heart tissues

using immunofluorescence co-localization of F4/80 and CD206 to further evaluate macrophage subsets. The relevant data were shown in Appendix Fig. S3F,G.

Comment 6: It is strongly recommended to have the manuscript proofread by a native English speaker.

Response: Thank you very much for your suggestion! We have proofread and polished the manuscript to enhance its readability.

Referee #2:

Comment 1: Based on authors data, EPA does prevent diabetic cardiomyopathy in vivo and in vitro experiments and HO-1 is involved in these phenotypes. However, HO-1 induction by other stimuli may lead to similar phenotype without EPA treatment. EPA may need HO-1 to show these beneficial phenotypes, but HO-1 induction (stronger) could be achieved by other stimuli and lead to better anti-DCM effects.

Response: Thank you very much for your insightful comments! I agree with your perspective. As you have noted, many studies have shown that HO-1 can exert antioxidant effects under various stimuli, thereby protecting against diabetic cardiomyopathy (DC). However, we demonstrated that EPA-induced HO-1 is a key gene for macrophage Mox polarization, and that EPA may exert a protective effect against DC by promoting the Mox phenotype of macrophages. Previous studies have identified HO-1 as a classic antioxidant gene. Our study is the first to report that HO-1 is a key gene in Mox polarized macrophages mediating protection against DC. Based on our findings, we hypothesized that other stimuli might similarly promote Mox polarized macrophages to protect DC, although such studies have not yet been reported. Future research in this area is warranted.

Comment 2: How does EPA modulate Mox polarization? What can be the possible molecular mechanism? Is there a particular membrane sensor for EPA and then transduce signaling to activate HO-1 in macrophages to polarize into Mox?

Response:

Thank you very much for your insightful comments! The direct connection between EPA and HO-1 in anti-DC effects was verified in further studies. We confirmed that the free fatty acid receptor 4 (FFA4, also known as GPR120) might be involved in EPA's elevation of HO-1.

G protein-coupled receptors (GPRs) are important signaling molecules for many cellular functions (Gether, 2000; Ulloa-Aguirre *et al*, 1999). It was reported that five orphan receptors, GPR40, GPR41, GPR43, GPR84, and GPR120, could be activated by free fatty acids (FFAs). Among these GPRs, GPR40 (Itoh *et al*, 2003) and GPR120 (Hirasawa *et al*, 2005) could interact with long-chain fatty acids (LCFAs). In particular, GPR120 mediates the involvement of LCFAs in regulating a variety of physiological processes, including the secretion of gut peptides, pancreatic function, food preference, appetite control, anti-inflammatory effects, adipogenesis, and insulin sensitization (Gribble *et al*, 2017; Liu *et al*, 2015; Mo *et al*, 2013; Tucker *et al*, 2014).

Therefore, GPR120 is considered an attractive potential target for treating metabolic dysfunctions such as type 2 diabetes, obesity, and inflammatory diseases.

Previous studies reported that GPR120 was highly expressed in M1 polarized macrophages (Oh *et al*, 2010), and GPR120-mediated signaling pathways could be activated by EPA in macrophages (Han *et al*, 2017). Thus, we hypothesized that EPA might function through activating macrophage GPR120 and/or GPR40.

Following the suggestion of referee #2, we employed Western blot to detect the GPR120 and GPR40 expression in EPA-treated M1 polarized RAW264.7 cells. We further investigated the effect of EPA on HO-1 activation after the application of the GPR120 antagonist AH7614 to validate our research hypothesis. The results showed that GPR120, rather than GPR40, might be involved in EPA's elevation of HO-1. The relevant results were displayed in Fig. 8 and Appendix Fig. S8.

Referee #3:

Major Comments:

Comment 1: The authors only use H&E staining to make general claims about immune cell infiltration; yet stain for F4/80 by confocal microscopy is used in other panels of the manuscript. Quantification of F4/80+ cells per area should be added to solidify the claims.

Response: Thank you very much for your reminder! We added immunohistochemical staining of F4/80 in heart tissue, and the result was displayed in Fig. EV2G,H.

Comment 2: iNOS staining as an M1 marker in Figure 2 I/J appears diffuse. Maybe use better representative images or more convincingly include an additional M1 surface marker to co-localize with F4/80?

Response: Thank you very much for your reminder! We added the immunofluorescence staining of another M1 marker (CD86) and F4/80 co-localization to make the results more convincing. The relevant results were showed in Fig. 2J,L.

Comment 3: The authors make use of the HL-1 cell line throughout the manuscript, yet previous work has demonstrated that HL-1 cells have impaired ability to shift metabolism in response to injurious stimuli (Kuznetsov *et al*. 2015 BBA). Please explain the rationale for using this cell type and/or include a statement in a limitations section?

Response: We appreciate your insightful reminder! Both H9c2 and HL-1 cell lines are immortalized cells with cardiac phenotype, and both are widely used for the analysis of cardiac ischemia-reperfusion injury and ischemic preconditioning (Pelloux *et al*, 2006). Indeed, previous work has shown that HL-1 cells have impaired ability to shift metabolism in response to injurious stimuli (Kuznetsov *et al*, 2015), which constituted a limitation of our study. Nonetheless, H9c2 cell is a permanent cell line derived from rat cardiac tissue (Hescheler *et al*, 1991). Given that the animal experiments involved a type 2 diabetes mouse model in this study, the mouse cardiomyocyte HL-1 cell line was selected for *in vitro* studies to align with the *in vivo* models. Additionally, it is

significant that HL-1 cells are a commonly used model for studying diabetic cardiomyopathy (Rodríguez-Calvo *et al*, 2019; Zhang *et al*, 2023a; Zhang *et al*, 2023b).

Comment 4: The authors make claims about the effect of EPA not impacting M2 polarization. Please include a description of the method how RAW cells are polarized to an M2 phenotype. Also please clarify the experimental set-up for this experiment. Furthermore, this data seems to be missing from the manuscript (i.e. figure S3 is missing).

Response: Thank you very much for your reminder! In the manuscript, we mainly confirmed that EPA has no significant effect on the M2 phenotype of M1 polarized macrophages (Appendix Fig. S3), rather than the effect of EPA on M2 polarization of macrophages. However, we greatly appreciate your attention. We clarified the method of M2 polarization in RAW264.7 cells and added an assessment of EPA's impact on macrophage M2 polarization. The results can be found in Appendix Fig. S4.

Comment 5: The authors need to confirm by RT-qPCR and Western blotting that Hmox1 siRNA effectively knocks down HMOX1 expression.

Response: Thank you very much for your reminder! The results were shown in Appendix Figure S6A,B.

Comment 6: Please clarify how cardiomyocyte size was measured as this is not clear from the methods or the figure.

Response: Thank you very much for your reminder! We used fluorescein-conjugated WGA staining (green fluorescence) to label the cardiomyocyte membranes and observe the structure of the cardiomyocytes. The complete green fluorescence contour represents the single cardiomyocyte. The cross-sectional areas of the cardiomyocytes were observed by WGA staining of the hearts from 8 mice per group, and evaluated by calculating the single myocyte cross-sectional areas measured by Image-Pro Plus software. We then statistically analyzed the mean cardiomyocyte areas, with all data normalized to the Ctrl group. Following your suggestion, we detailed the measurement methods in the "Methods-Assessment of cardiac pathology" section of our manuscript.

Comment 7: The authors report decreased heart size, but also concurrent increases in body weight. In this case it would be appropriate to normalize heart mass to tibial length or some other standard which is unchanged in the DM mice.

Response: Thank you very much for your reminder! We normalized heart mass to tibia length, and the results were displayed in Fig. EV2B.

Comment 8: Please clarify whether perivascular Masson's trichrome staining was included or excluded for quantification.

Response: Thank you very much for your reminder! The perivascular Masson's trichrome staining was included for quantification.

Minor Comments:

Comment 1: Be consistent with references to supplemental figures throughout the text.

Response: Thank you very much for your reminder! We double-checked the manuscript so that references to supplemental figures throughout the text are consistent. The “**MANUSCRIPT PREPARATION**” section of the journal states that EMBO Press strongly encourages authors to select a limited number (typically 5) of supplementary figures for inclusion in the article proper as Expanded View figures in order to improve their accessibility, visibility and utility. Any extra figures that are not promoted to the Expanded View should be included in a 'traditional' supplementary PDF (along with supplementary text and tables) now called the Appendix. Therefore, we showed the main supplementary figures in Figures EV1-EV5, and the remaining extended data in Appendix Figures S1-S8.

Comment 2: Labels should be included on the GTT and ITT graphs in supplementary figure.

Response: Thank you very much for your reminder! We labeled the GTT and ITT graphs in Fig. EV1D,F.

Comment 3: Are other NOX isoforms (in addition to NOX4) changed in the HL-1 co-culture experiments?

Response: Thank you very much for your reminder! The NOX2 protein level was measured in the HL-1 co-culture experiments, and the results were displayed in Fig. 3I.

We look forward to working together with you to ensure that this manuscript achieves the best quality and content. If you have any further suggestions or comments, please don't hesitate to let me know. Thank you for your patience and support!

Best regards,

Hao Wu, M.D., Ph.D.

Professor

Department of Nutrition and Food Hygiene,
Shandong University School of Public Health
44 Wenhuaaxi Rd., Jinan, Shandong 250012, China

Tel.: +86 531 88382135

Fax: +86 531 88382129

Email: hwu@sdu.edu.cn

ORCID: 0000-0001-5738-4015

References

- Barry JC, Shakibakho S, Durrer C, Simtchouk S, Jawanda KK, Cheung ST, Mui AL, Little JP (2016) Hyporesponsiveness to the anti-inflammatory action of interleukin-10 in type 2 diabetes. *Scientific reports* 6: 21244
- Chawla A, Nguyen KD, Goh YP (2011) Macrophage-mediated inflammation in metabolic disease. *Nature reviews Immunology* 11: 738-749
- Feng R, Luo C, Li C, Du S, Okekunle AP, Li Y, Chen Y, Zi T, Niu Y (2017) Free fatty acids profile among lean, overweight and obese non-alcoholic fatty liver disease patients: a case - control study. *Lipids in health and disease* 16: 165
- Gether U (2000) Uncovering molecular mechanisms involved in activation of G protein-coupled receptors. *Endocrine reviews* 21: 90-113
- Gribble FM, Diakogiannaki E, Reimann F (2017) Gut Hormone Regulation and Secretion via FFA1 and FFA4. *Handbook of experimental pharmacology* 236: 181-203
- Han L, Song S, Niu Y, Meng M, Wang C (2017) Eicosapentaenoic Acid (EPA) Induced Macrophages Activation through GPR120-Mediated Raf-ERK1/2-IKK β -NF- κ B p65 Signaling Pathways. *Nutrients* 9
- Hescheler J, Meyer R, Plant S, Krautwurst D, Rosenthal W, Schultz G (1991) Morphological, biochemical, and electrophysiological characterization of a clonal cell (H9c2) line from rat heart. *Circulation research* 69: 1476-1486
- Hirasawa A, Tsumaya K, Awaji T, Katsuma S, Adachi T, Yamada M, Sugimoto Y, Miyazaki S, Tsujimoto G (2005) Free fatty acids regulate gut incretin glucagon-like peptide-1 secretion through GPR120. *Nature medicine* 11: 90-94
- Itoh Y, Kawamata Y, Harada M, Kobayashi M, Fujii R, Fukusumi S, Ogi K, Hosoya M, Tanaka Y, Uejima H *et al* (2003) Free fatty acids regulate insulin secretion from pancreatic beta cells through GPR40. *Nature* 422: 173-176
- Kuznetsov AV, Javadov S, Sickinger S, Frotschnig S, Grimm M (2015) H9c2 and HL-1 cells demonstrate distinct features of energy metabolism, mitochondrial function and sensitivity to hypoxia-reoxygenation. *Biochimica et biophysica acta* 1853: 276-284
- Liu HD, Wang WB, Xu ZG, Liu CH, He DF, Du LP, Li MY, Yu X, Sun JP (2015) FFA4 receptor (GPR120): A hot target for the development of anti-diabetic therapies. *European journal of pharmacology* 763: 160-168
- Liu Y, Tian Y, Dai X, Liu T, Zhang Y, Wang S, Shi H, Yin J, Xu T, Zhu R *et al* (2023) Lycopene ameliorates islet function and down-regulates the TLR4/MyD88/NF- κ B pathway in diabetic mice and Min6 cells. *Food & function* 14: 5090-5104
- Mo XL, Wei HK, Peng J, Tao YX (2013) Free fatty acid receptor GPR120 and pathogenesis of obesity and type 2 diabetes mellitus. *Progress in molecular biology and translational science* 114: 251-276
- Moganti K, Li F, Schmuttermaier C, Riemann S, Klüter H, Gratchev A, Harmsen MC, Kzhyshkowska J (2017) Hyperglycemia induces mixed M1/M2 cytokine profile in primary human monocyte-derived macrophages. *Immunobiology* 222: 952-959
- Morey M, O'Gaora P, Pandit A, H elary C (2019) Hyperglycemia acts in synergy with

- hypoxia to maintain the pro-inflammatory phenotype of macrophages. *PloS one* 14: e0220577
- Oh DY, Talukdar S, Bae EJ, Imamura T, Morinaga H, Fan W, Li P, Lu WJ, Watkins SM, Olefsky JM (2010) GPR120 is an omega-3 fatty acid receptor mediating potent anti-inflammatory and insulin-sensitizing effects. *Cell* 142: 687-698
- Pavlou S, Lindsay J, Ingram R, Xu H, Chen M (2018) Sustained high glucose exposure sensitizes macrophage responses to cytokine stimuli but reduces their phagocytic activity. *BMC immunology* 19: 24
- Pelloux S, Robillard J, Ferrera R, Bilbaut A, Ojeda C, Saks V, Ovize M, Tournier Y (2006) Non-beating HL-1 cells for confocal microscopy: application to mitochondrial functions during cardiac preconditioning. *Progress in biophysics and molecular biology* 90: 270-298
- Rao Z, Sun J, Pan X, Chen Z, Sun H, Zhang P, Gao M, Ding Z, Liu C (2017) Hyperglycemia Aggravates Hepatic Ischemia and Reperfusion Injury by Inhibiting Liver-Resident Macrophage M2 Polarization via C/EBP Homologous Protein-Mediated Endoplasmic Reticulum Stress. *Frontiers in immunology* 8: 1299
- Rendra E, Riabov V, Mossel DM, Sevastyanova T, Harmsen MC, Kzhyskowska J (2019) Reactive oxygen species (ROS) in macrophage activation and function in diabetes. *Immunobiology* 224: 242-253
- Rodríguez-Calvo R, Girona J, Rodríguez M, Samino S, Barroso E, de Gonzalo-Calvo D, Guaita-Esteruelas S, Heras M, van der Meer RW, Lamb HJ *et al* (2019) Fatty acid binding protein 4 (FABP4) as a potential biomarker reflecting myocardial lipid storage in type 2 diabetes. *Metabolism: clinical and experimental* 96: 12-21
- Thapa B, Lee K (2019) Metabolic influence on macrophage polarization and pathogenesis. *BMB reports* 52: 360-372
- Torres-Castro I, Arroyo-Camarena Ú D, Martínez-Reyes CP, Gómez-Arauz AY, Dueñas-Andrade Y, Hernández-Ruiz J, Béjar YL, Zaga-Clavellina V, Morales-Montor J, Terrazas LI *et al* (2016) Human monocytes and macrophages undergo M1-type inflammatory polarization in response to high levels of glucose. *Immunology letters* 176: 81-89
- Tucker RM, Mattes RD, Running CA (2014) Mechanisms and effects of "fat taste" in humans. *BioFactors (Oxford, England)* 40: 313-326
- Ulloa-Aguirre A, Stanislaus D, Janovick JA, Conn PM (1999) Structure-activity relationships of G protein-coupled receptors. *Archives of medical research* 30: 420-435
- Weisberg SP, McCann D, Desai M, Rosenbaum M, Leibel RL, Ferrante AW, Jr. (2003) Obesity is associated with macrophage accumulation in adipose tissue. *The Journal of clinical investigation* 112: 1796-1808
- Wen Y, Gu J, Li SL, Reddy MA, Natarajan R, Nadler JL (2006) Elevated glucose and diabetes promote interleukin-12 cytokine gene expression in mouse macrophages. *Endocrinology* 147: 2518-2525
- Zhang S, Wang M, Li H, Li Q, Liu N, Dong S, Zhao Y, Pang K, Huang J, Ren C *et al* (2023a) Exogenous H(2) S promotes ubiquitin-mediated degradation of SREBP1

to alleviate diabetic cardiomyopathy via SYVN1 S-sulfhydration. *Journal of cachexia, sarcopenia and muscle* 14: 2719-2732

Zhang W, Lu J, Wang Y, Sun P, Gao T, Xu N, Zhang Y, Xie W (2023b) Canagliflozin Attenuates Lipotoxicity in Cardiomyocytes by Inhibiting Inflammation and Ferroptosis through Activating AMPK Pathway. *International journal of molecular sciences* 24

Zhou Q, Wang Y, Lu Z, Wang B, Li L, You M, Wang L, Cao T, Zhao Y, Li Q *et al* (2023) Mitochondrial dysfunction caused by SIRT3 inhibition drives proinflammatory macrophage polarization in obesity. *Obesity (Silver Spring, Md)* 31: 1050-1063

Dear Hao,

Thank you for submitting your revised manuscript. It has now been seen by two of the original referees.

I apologize for this unusual delay in getting back to you. It took longer than anticipated to receive the referee reports.

As you can see, the referees find that the study is significantly improved during revision and recommend publication. However, I need you to address the points below before I can accept the manuscript.

- Please move the Disclosure and competing interests statement after the Acknowledgements section.
- Please remove the Author contributions section from the manuscript text.
- We note the following regarding the Appendix file
 - o Please remove the first page with the title page with author names and affiliations.
 - o Table of Contents should have each item (figures and tables) listed with its page number
 - o Nomenclature of the items needs to be corrected as Appendix Figure S1, Appendix Table S1, etc.
- Please upload source data as one file per figure.
- Please remove "BioRender was used to prepare the synopsis of this study." from the Acknowledgement section. Instead, please add this information the Methods section as a separate subheading at the very end as follows: "Graphics. BioRender was used to prepare the synopsis of this study."
- Table 1 should be placed between main and EV figure legends.
- Please provide specific URLs for datasets PRJNA1007889 and PRJNA1007927 in the data availability statement.
- Our production/data editors have asked you to clarify several points in the figure legends:
 1. Please note that the legends for figures EV 3f-g is not provided in the sequential manner (legend for figure 3g is provided before legend of figure 3f). This needs to be rectified.
 2. Please note that the legends for figures 2j-k is not provided in the sequential manner (legend for figure 2k is provided before legend of figure 2j). This needs to be rectified."
 3. Please note that the exact p values are not provided in the legends of figures 1b-c, f-h; 2a-f, h, k, l; 3b-i, k; 4b-g; 5b-f, h-k; 6b-f; 7c-d, g-h, j; 8a-d; EV 1b-g; EV 2b, d, f, h, j, l; EV 3a-f; EV 5b-i.
 4. Please note that in figure EV 2b; there is a mismatch between the annotated p values in the figure legend and the annotated p values in the figure file that should be corrected.
 5. Please note that the white arrows are not defined in the legend of figure 2i, j; 7e-f. This needs to be rectified.

Thank you again for giving us to consider your manuscript for EMBO Reports, I look forward to your minor revision.

Kind regards,

Deniz

--

Deniz Senyilmaz Tiebe, PhD
Senior Scientific Editor
EMBO Reports

Referee #2:

Authors addressed all aspects of my previous concerns and questions.

Referee #3:

The authors addressed this reviewer's critiques.
No more comments.

All editorial and formatting issues were resolved by the authors.

Prof. Hao Wu
Shandong University School of Public Health
China

Dear Hao,

Thank you for submitting your revised manuscript. I have now looked at everything and all is fine. Therefore, I am very pleased to accept your manuscript for publication in EMBO Reports.

Congratulations on a nice work!

Kind regards,

Deniz

--

Deniz Senyilmaz Tiebe, PhD
Senior Scientific Editor
EMBO Reports
